# The Community Inversion Framework v1.0: a unified system for atmospheric inversion studies

Antoine Berchet[1,*], Espen Sollum[2], Rona L. Thompson[2], Isabelle Pison[1], Joël Thanwerdas[1], Grégoire Broquet[1], Frédéric Chevallier[1], Tuula Aalto[3], Adrien Berchet[14], Peter Bergamaschi[4], Dominik Brunner[5], Richard Engelen[6], Audrey Fortems-Cheiney[1], Christoph Gerbig[7], Christine D. Groot Zwaaftink[2], Jean-Matthieu Haussaire[5], Stephan Henne[5], Sander Houweling[8], Ute Kartens[9], Werner L. Kutsch[10], Ingrid T. Luijkx[11], Guillaume Monteil[9], Paul I. Palmer[12], Jacob C. A. van Peet[8], Wouter Peters[11,13], Philippe Peylin[1], Elise Potier[1], Christian Rödenbeck[7], Marielle Saunois[1], Marko Scholze[9], Aki Tsuruta[3] and Yuanhong Zhao[1]

[1]Laboratoire des Sciences du Climat et de l'Environnement, CEA-CNRS-UVSQ, Gif-sur-Yvette, France
[2]Norwegian Institute for Air Research (NILU), Kjeller, Norway
[3]Finnish Meteorological Institute (FMI), Helsinki, Finland
[4]European Commission Joint Research Centre, Ispra (Va), Italy
[5]Swiss Federal Laboratories for Materials Science and Technology (Empa), Dübendorf, Switzerland
[6]European Centre for Medium-Range Weather Forecasts, Reading, RG2 9AX, UK
[7]Max Planck Institute for Biogeochemistry, Jena, Germany
[8]Vrije Universiteit Amsterdam, Department of Earth Sciences, Earth and Climate Cluster, Amsterdam, the Netherlands
[9]Dep. of Physical Geography and Ecosystem Science, Lund University, Sweden
[10]Integrated Carbon Observation System (ICOS-ERIC), Helsinki, Finland
[11]Meteorology and Air Quality Group, Wageningen University and Research, Wageningen, the Netherlands
[12]School of GeoSciences, University of Edinburgh, Edinburgh, EH9 3FF, UK
[13]Centre for Isotope Research, University of Groningen, Groningen, the Netherlands
[14]Institut Pprime (UPR 3346, CNRS-Université de Poitiers-ENSMA), Bat H2, 11 Boulevard Marie et Pierre Curie, TSA 51124, 86073, Poitiers Cedex 9, France

**Correspondence:** antoine.berchet@lsce.ipsl.fr

**Abstract.** Atmospheric inversion approaches are expected to play a critical role in future observation-based monitoring systems for surface fluxes of greenhouse gas (GHG), pollutants and other trace gases. In the past decade, the research community has developed various inversion softwares, mainly using variational or ensemble Bayesian optimization methods, with various assumptions on uncertainty structures and prior
5   information and with various atmospheric chemistry-transport models. Each of them can assimilate some or all of the available observation streams for its domain area of interest: flask samples, in-situ measurements or satellite observations. Although referenced in peer-reviewed publications and usually accessible across the research community, most systems are not at the level of transparency, flexibility and accessibility needed to provide the scientific community and policy makers with a comprehensive and robust view of
10   the uncertainties associated with the inverse estimation of GHG and reactive species fluxes. Furthermore,

their development, usually carried out by individual research institutes, may in the future not keep pace with the increasing scientific needs and technical possibilities. We present here a *Community Inversion Framework* (CIF) to help rationalize development efforts and leverage the strengths of individual inversion systems into a comprehensive framework. The CIF is primarily a programming protocol to allow various inversion bricks to be exchanged among researchers. In practice, the ensemble of bricks makes a flexible, transparent and open-source python-based tool to estimate the fluxes of various GHGs and reactive species both at the global and regional scales. It will allow running different atmospheric transport models, different observation streams and different data assimilation approaches. This adaptability will allow a comprehensive assessment of uncertainty in a fully consistent framework. We present here the main structure and functionalities of the system, and demonstrate how it operates in a simple academic case.

## 1  Introduction

The role of greenhouse gases (GHGs) in climate change has motivated an exceptional effort over the last couple of decades to densify the observations of GHGs around the world (Ciais et al., 2014): from the ground, e.g., with the European Integrated Carbon Observation System (ICOS, https://www.icos-cp.eu/), from mobile platforms (e.g., from aircrafts, or balloons equipped with Aircore sampling; Filges et al., 2016; Karion et al., 2010), and from space (e.g., Crisp et al., 2018; Janssens-Maenhout et al., 2020), despite occasional budgetary difficulties (Houweling et al., 2012). These observations quantify the effect of exchange between the surface and the atmosphere on GHG concentrations (e.g., Ramonet et al., 2020) and can thus be used to determine the surface fluxes of GHGs through the inversion of atmospheric chemistry and transport (e.g., Peylin et al., 2013; Houweling et al., 2017). Alongside improved observation capabilities, national and international initiatives pave the way towards an operational use of atmospheric inversions to support emissions reporting to the United Nations Framework Convention on Climate Change (UNFCCC; e.g., Say et al., 2016; Henne et al., 2016; Bergamaschi et al., 2018a; Janssens-Maenhout et al., 2020, or the EU projects CHE – $CO_2$ Human emissions; che-project.eu – or VERIFY – verify.lsce.ipsl.fr).

  In the past, research groups have developed various atmospheric inversion systems based on different techniques and atmospheric transport models, targeting specific trace gases or types of observations, and at various spatial and temporal scales, according to the particular scientific objectives of the study. All these systems have their own strengths and weaknesses and help explore the range of systematic uncertainty in the surface to atmosphere fluxes. Inter-comparison exercises are regularly conducted to assess the strengths and weaknesses of various inversion systems (e.g., Gurney et al., 2003; Peylin et al., 2013; Locatelli et al., 2013; Babenhauserheide et al., 2015; Brunner et al., 2017; Bergamaschi et al., 2018b; Chevallier et al., 2019; Crowell et al., 2019; Monteil et al., 2019; Schuh et al., 2019; Saunois et al., 2020). Inter-comparisons also provide an assessment of the systematic uncertainty on final flux estimates induced by the variety of options

and choices in different inversion systems. However, although the inversion systems are referenced in peer-reviewed literature, and are usually accessible to the research community, they are typically not at the level of transparency, documentation, flexibility and accessibility required to provide both the scientific community and policy makers with a comprehensive and robust view of the uncertainties associated with the inverse estimation of trace gases (primarily GHGs and reactive species) fluxes. In particular, the differences between inversion systems (such as the atmospheric transport model, prior and observation space uncertainties, and inversion algorithm) make comparing their results particularly challenging, even when they are applied to the same problem. Moreover, research inversion systems are so far not ready for operational use, and their development, usually carried out by individual research institutes or limited consortia, may not keep pace with the scientific and technical needs to come, such as those linked to the increasing availability of high resolution satellite GHG and reactive species observations (Janssens-Maenhout et al., 2020). A unified system, as a community platform running multiple transport models, with diverse inversion methods, would provide new possibilities to effectively and comprehensively assess GHG and various reactive species budgets, trends, and their uncertainties and quantify limitations and development needs related to different approaches, all which is needed in order to properly support emission reporting. Collaborative efforts towards unified systems are already happening in other data assimilation communities, with, e.g., the Object-Oriented Prediction System (OOPS; coordinated by the European Centre for Medium-range Weather Forecast, UK), or the Joint Effort for Data Integration (led by UCAR/JCSDA; www.jcsda.org/jcsda-project-jedi). The Data Assimilation Research Testbed (DART; Anderson et al., 2009) is also an example of collective effort proposing common data assimilation scripts for diverse applications (e.g., Earth system, or reactive species inversions; Gaubert et al. 2020). The *Community Inversion Framework* (CIF) is an initiative by members of the GHG atmospheric inversion community to bring together the different inversion systems used in the community, and is supported by the European Commission H2020 project VERIFY. The CIF will also support operational applications of atmospheric inversions in the CoCO2 project (coco2-project.eu), that will design an operational inversion system based on OOPS and interfaced with the research community through the CIF.

Despite their differences in methodology, application and implementation, almost all inversion systems rely on the same conceptual and practical bases: in particular, they use model-observation mismatches in a statistical optimization framework (most of the time based on Bayes' Theorem), and numerical atmospheric tracer transport and chemistry models to simulate mixing ratios of GHGs and trace gases based on surface fluxes. The objectives of the CIF are to develop a consistent input/model interface, to pool development efforts, and to have an inversion tool that is well-documented, open-source, and ready for implementation in an operational framework. The CIF is designed to be a flexible and transparent tool to estimate the fluxes of different GHGs (e.g. carbon dioxide $CO_2$, methane $CH_4$, nitrous oxide $N_2O$, or halocarbons) and other species, such as reactive species (e.g. CO, $NO_2$, HCHO), based on atmospheric measurements. In particular, although primarily designed for GHGs applications, the CIF is based on a general structure that will

allow applications to air quality data assimilation. It is also designed to run at any spatial and temporal scales and with different atmospheric (chemistry-)transport models (global and regional, Eulerian and Lagrangian), with various observation data streams (ground-based, remote sensing, etc.), and a variety of data assimilation techniques (variational, analytical, ensemble methods, etc.). It will be possible to run it on mul-
5 tiple computing environments and corresponding set-ups and tutorials will be well documented. Community development will help in tackling the challenges in set-up and running, and accelerate adoption of the tool into wider use. One of the main foreseen advantages of the CIF is the capability to quantify and compare the errors due to the modelling of atmospheric transport and the errors due to the choice of a given inversion approach and set-up to solve a specific problem, in a fully consistent framework. The CIF will provide a
10 common platform for quickly developing and testing new inversion techniques with several transport models, and it is hoped that with the combined community effort, it will be continuously improved and revised, keeping it state-of-the-art.

In the present paper, we lay out the basis of the CIF, giving details on its underlying principles and overall implementation. The proof-of-concept focuses on the implementation of several inversion methods,
illustrated with a test case. We will dedicate a future paper to the evaluation of the system on a real-life problem with a number of interfaced atmospheric (chemistry-)transport models. At the time of writing the present article, the following models are interfaced with the CIF: the Global Circulation Models LMDZ (Chevallier et al., 2005) and TM5 (Krol et al., 2005; van der Laan-Luijkx et al., 2017), the regional chemistry-transport Eulerian model CHIMERE (Fortems-Cheiney et al., 2021) and the Lagrangian particle dispersion
models FLEXPART (Pisso et al., 2019) and STILT (Trusilova et al., 2010). For the sake of simplicity, we refer to all types of (chemistry-)transport models generically as CTMs in the following. In Section 2, we describe the general theoretical framework for atmospheric inversions and how the CIF will include the theory in a flexible and general way. In Section 3, the practical implementation of the general design rules is explained, with details on the python implementation of the CIF. In Section 4, we demonstrate the capabilities
of the CIF in a simple test case, applying various inversion techniques in parallel.

## 2 General principle

The version of the CIF presented here is implemented around Bayesian data assimilation methods with Gaussian assumptions, which constitute the main framework used in the atmospheric inversion systems for GHG fluxes and other trace gases (e.g., Enting, 2002; Bocquet et al., 2015). However, some studies have
30 proposed possible extensions to more general probability density functions beyond the classical Gaussian case (e.g., truncated Gaussian densities, log-normal distributions, etc.; Michalak and Kitanidis, 2005; Bergamaschi et al., 2010; Miller et al., 2014; Zammit-Mangion et al., 2015; Lunt et al., 2016; Miller et al., 2019). Therefore, we propose here a general and flexible structure for our system that will be independent of limiting

assumptions, as described in Sect 2.3, to allow future extensions to more general theoretical frameworks. In the following, mathematical formulas are written following notations based on Ide et al. (1997) and Rayner et al. (2019). We present the theoretical basis and several inversion methods that are implemented in the CIF as demonstrators.

## 2.1   General Bayesian data assimilation framework

The Bayesian approach consists in estimating the following conditional probability density function (pdf):

$$p^{\mathrm{a}}(\mathbf{x}) = p(\mathbf{x} \mid \mathbf{y}^{\mathrm{o}} - \mathcal{H}(\mathbf{x}^{\mathrm{b}}), \mathbf{x}^{\mathrm{b}}) \propto p(\mathbf{y}^{\mathrm{o}} - \mathcal{H}(\mathbf{x}) \mid \mathbf{x})\, p^{\mathrm{b}}(\mathbf{x}) \tag{1}$$

with $\mathbf{x}$ the target vector, $p^{\mathrm{a}}(\mathbf{x})$ the posterior distribution of the target vector, $p^{\mathrm{b}}(\mathbf{x})$ the prior knowledge of the target vector, characterized by its mode $\mathbf{x}^{\mathrm{b}}$, $\mathbf{y}^{\mathrm{o}}$ the observation vector gathering all observations imple-

mented in the inversion and $\mathcal{H}$ the observation operator linking the target vector to the observation vector. In the following, we also refer to $\mathcal{X}$ and $\mathcal{Y}$ as the target and observation spaces, respectively, from where the target and observation vectors are sampled. Classically, for atmospheric inversions, the observation vector $\mathbf{y}^{\mathrm{o}}$ includes ground-based measurements of trace gases mixing ratios on fixed or mobile platforms, and remote sensing observations such as satellite observations. The target vector $\mathbf{x}$ includes the variables to be optimized

by the inversion; it includes the main variables of interest, such as the surface fluxes, but also variables related to atmospheric chemical sources and sinks, background concentrations in the case of limited-area transport models, model parameters, etc., which are required to make the inversion physically consistent. The observation operator $\mathcal{H}$ mainly includes the computation of atmospheric transport and chemistry (if relevant) by numerical (chemistry-)transport models. These can be of various types: e.g., global transport models (e.g.,

LMDZ, Chevallier et al. 2010; TM5, Houweling et al. 2014; GEOS-Chem, van der Laan-Luijkx et al. 2017; Liu et al. 2015; Palmer et al. 2019; Feng et al. 2017; NICAM, Niwa et al. 2017), regional Eulerian chemistry-transport models (e.g., CHIMERE, Broquet et al. 2011; Fortems-Cheiney et al. 2021; WRF-CHEM, Zheng et al. 2018; COSMO-GHG, Kuhlmann et al. 2019; LOTOS-EUROS, Curier et al. 2012) or Lagrangian particle dispersion models (e.g., FLEXPART, Thompson and Stohl 2014; STILT, Bagley et al. 2017; Brioude

et al. 2013; Trusilova et al. 2010). It also includes pre- and post-processing operations required to project the target vector to a format compatible with the model input and the model outputs to the observation vector; these operations can be the applications of e.g., averaging kernels in the case of satellite operations, or interpolation of the target vector to higher resolution model inputs.

As errors in inversion systems come from a large variety of independent causes superimposing on each

other, it is often assumed that the most relevant way of representing the distributions in Eq. (1) is to assume prior and observation spaces to be normal distributions, noted $\mathcal{N}(\cdot, \cdot)$ below, the first argument representing the average of the distribution and the second argument the covariance matrix. In addition, when assuming that the distributions in the state vector space and the observation space are independent from each other, and

that errors in the observation and the state vector spaces have Gaussian, unbiased distributions, it is possible mathematically derive the Bayes theorem and to represent the distributions of Eq. (1) as follows:

$$
\begin{cases}
p^{\mathrm{b}}(\mathbf{x}) & \sim \quad \mathcal{N}(\mathbf{x}^{\mathrm{b}},\, \mathbf{B}) \\
p\left(\mathbf{y}^{\mathrm{o}} - \mathcal{H}(\mathbf{x}^{\mathrm{b}})\right) & \sim \quad \mathcal{N}(\mathbf{0},\, \mathbf{R}) \\
p^{\mathrm{a}}(\mathbf{x}) & \sim \quad \mathcal{N}(\mathbf{x}^{\mathrm{a}},\, \mathbf{A})
\end{cases}
\tag{2}
$$

with $\mathbf{B}$ and $\mathbf{A}$ the prior and posterior covariance matrix of uncertainties in the target vector, $\mathbf{x}^{\mathrm{b}}$ and $\mathbf{x}^{\mathrm{a}}$ the prior and posterior target vectors and $\mathbf{R}$ the covariance matrix of uncertainties in the observation vector and the observation operator.

The assumption that errors are unbiased, which makes it possible to write normal distributions in Eq. (1) with means $\mathbf{x}^{\mathrm{b}}$, $\mathbf{0}$ and $\mathbf{x}^{\mathrm{a}}$ respectively, is needed to simplify the formulation of the Bayesian problem in atmospheric inversions. Neglecting error biases have known impacts on inversion results (e.g., Masarie et al., 2011); they can be accounted for online as an unknown to be solved by the inversion (e.g., Zammit-Mangion et al., 2021), but are often treated offline from the inversion, either through pre-processing of inputs or post-processing of outputs.

## 2.2 Computation modes in the CIF

The present version of the CIF includes three main categories of inversion methods: 1) analytical, i.e. algebraic solution of the unbiased Gaussian Bayesian problem, 2) ensemble methods with the Ensemble Square Root Filter (EnSRF), and 3) variational with two examples of minimizing algorithms (M1QN3 and CONGRAD). Other types of data assimilation methods (e.g. direct sampling of probability density functions through Monte Carlo approaches) are also used by the community. The choice of implementing the three aforementioned methods first is motivated by their dominant use, and because these three use very different approaches for solving the Bayesian inversion problem: that is, with/without random sampling of probability distributions, and with/without the use of the adjoint of the observation operator. The adjoint of the observation operator, noted $\mathcal{H}^*$, is built following the mathematical definition of the adjoint; heuristically, it operates backwards compared to the observation operator (e.g., Errico, 1997) in the sense that it determines the sensitivity to inputs (e.g. fluxes) given an incremental perturbation to outputs (e.g. concentrations). In addition to the mentioned data assimilation methods, the CIF also includes the possibility to run forward simulations and to test the adjoint and the tangent linear of the observation operator for given inversion configurations. In the following we call all inversion methods and other types of computation in the CIF "computation modes". With these computation modes implemented in a flexible and general manner, it is anticipated that other inversion methods could be easily added to the CIF in the future (see Sect. 2.3).

### 2.2.1 Data assimilation methods

**Analytical inversions**

Analytical inversions compute the algebraic solution of the Gaussian Bayesian problem when it is linear and are used extensively at all scales (e.g., Stohl et al., 2009; Turner and Jacob, 2015; Kopacz et al., 2009; Bousquet et al., 2011; Wang et al., 2018; Palmer et al., 2006). When the observation operator is linear, $\mathcal{H}$ equals its Jacobian matrix $\mathbf{H}$, and conversely its adjoint $\mathcal{H}^*$ is the transpose of the Jacobian $\mathbf{H}^\mathrm{T}$. In that case, $\mathbf{x}^\mathrm{a}$ and $\mathbf{A}$ can be explicitly written as matrix products. There are two equivalent formulations of the matrix products for the solution of the problem (e.g., Tarantola and Valette, 1982):

$$\left\{ \begin{array}{lll} \mathbf{x}^\mathrm{a} & = & \mathbf{x}^\mathrm{b} + \mathbf{K}(\mathbf{y}^\mathrm{o} - \mathbf{H}\mathbf{x}^\mathrm{b}) \\ \mathbf{A} & = & \mathbf{B} - \mathbf{K}\mathbf{H}\mathbf{B} \end{array} \right. \text{ or } \left\{ \begin{array}{lll} \mathbf{x}^\mathrm{a} & = & \mathbf{x}^\mathrm{b} + (\mathbf{H}^\mathrm{T}\mathbf{R}^{-1}\mathbf{H} + \mathbf{B}^{-1})^{-1}\mathbf{H}^\mathrm{T}\mathbf{R}^{-1}(\mathbf{y}^\mathrm{o} - \mathbf{H}\mathbf{x}^\mathrm{b}) \\ \mathbf{A} & = & (\mathbf{H}^\mathrm{T}\mathbf{R}^{-1}\mathbf{H} + \mathbf{B}^{-1})^{-1} \end{array} \right. \tag{3}$$

with $\mathbf{K}$ the Kalman gain matrix: $\mathbf{K} = \mathbf{B}\mathbf{H}^\mathrm{T}(\mathbf{R} + \mathbf{H}\mathbf{B}\mathbf{H}^\mathrm{T})^{-1}$

Analytical inversions can also be used on slightly non-linear problems, by linearizing the observation operator around a given reference point using the tangent linear of the observation operator. It formulates as follows:

$$\mathcal{H}(\mathbf{x}^\mathrm{b} + \delta\mathbf{x}) \approx \mathcal{H}(\mathbf{x}^\mathrm{b}) + d\mathcal{H}_{\mathbf{x}^\mathrm{b}}(\delta\mathbf{x}) = \mathcal{H}(\mathbf{x}^\mathrm{b}) + \mathbf{H}_{\mathbf{x}^\mathrm{b}}\delta\mathbf{x} \tag{4}$$

with $\delta\mathbf{x}$ a small deviation from $\mathbf{x}^\mathrm{b}$ within a domain where the linear assumption is valid, $d\mathcal{H}_{\mathbf{x}^\mathrm{b}}$ the tangent-linear of $\mathcal{H}$ at $\mathbf{x}^\mathrm{b}$ and $\mathbf{H}_{\mathbf{x}^\mathrm{b}}$ the Jacobian matrix of $\mathcal{H}$ at $\mathbf{x}^\mathrm{b}$.

Then Eq. (3) can be easily adapted by replacing $(\mathbf{y}^\mathrm{o} - \mathbf{H}\mathbf{x}^\mathrm{b})$ by $(\mathbf{y}^\mathrm{o} - \mathcal{H}(\mathbf{x}^\mathrm{b}))$ and $\mathbf{H}$ by $\mathbf{H}_{\mathbf{x}^\mathrm{b}}$.

The computation of an analytical inversion faces two main computational limitations. First, the matrix $\mathbf{H}$ representing the observation operator $\mathcal{H}$ must be built explicitly. This can be done either column by column, in the so-called response function method, or row by row, in the so-called footprint method. The two approaches require $\dim(\mathcal{X})$, the dimension of the target space and $\dim(\mathcal{Y})$, the dimension of the observation space, independent simulations respectively. In the response function method, each column is built by computing $\{d\mathcal{H}_{\mathbf{x}^\mathrm{b}}(\delta\mathbf{x}_i) \setminus \forall \delta\mathbf{x}_i \in \mathcal{B}_\chi\}$ with $\mathcal{B}_\chi$ the canonical basis of the target space. For a given increment $\delta\mathbf{x}_i$, the corresponding column gives the sensitivity of observations to changes in the corresponding component of the target space. In the footprint method, each row is built by computing $\{\mathcal{H}^*_{\mathbf{x}^\mathrm{b}}(\delta\mathbf{y}_i) \setminus \forall \delta\mathbf{y}_i \in \mathcal{B}_\mathcal{Y}\}$ with $\mathcal{B}_\mathcal{Y}$ the canonical basis of the observation space. For a given perturbation of $\delta\mathbf{y}_i$ of a component of the observation vector, the corresponding row of $\mathbf{H}$ gives the sensitivity of the inputs to that perturbation.

Depending on the number of available observations or the size of the target vector, one of the two is preferred to limit the number of observation operator computations to be carried out explicitly. When the dimension of the target vector is relatively small, the response function is generally preferred, and conversely,

when the observation vector is small, the footprint approach is preferred. The type of transport model used to compute the matrix $\mathbf{H}$ also plays a role in the choice of the approach: for Eulerian models, the response function approach is preferred for multiple reasons: (i) their adjoint is often much more costly than their forward, (ii) the adjoint may not be available for some models or is difficult to generate, and (iii) the computation time of the forward is constant no matter how numerous the observations; for Lagrangian models, the footprint approach is preferred as they often compute backward transport simulations for each observation, allowing a straightforward computation of the adjoint (Seibert and Frank, 2004). In both cases, the explicit construction of the matrix $\mathbf{H}$ requires numerous independent simulations, which can be an insurmountable computational challenge.

The second obstacle consists in that the computation of the Kalman gain matrix in Eq. (3) (left) requires inverting a matrix of the dimension of the observation space, $\dim(\mathcal{Y})$, while for the other formulation (Eq. (3) right) the matrix is of dimension $\dim(\mathcal{X})$, the dimension of the target space. If the dimensions of both the observation and the target spaces are very high, as in many inversion applications, the explicit computation of Eq. (3) with matrix products and inverses is not computationally feasible. For this reason, smart adaptations of the inversion framework (including approximations and numerical solvers) are often necessary to tackle problems even when they are linear; in the following, we choose to elaborate on some of the most frequent approaches used in the atmospheric inversion community: the variational approach and one ensemble method, the Ensemble Square Root Filter (EnSRF). Less frequently, intermediate adaptations of the analytical inversion also include sequential applications (e.g., Michalak, 2008; Bruhwiler et al., 2005; Brunner et al., 2012), that are a compromise between tackling the above-mentioned computational obstacles while maintaining the simplicity of the analytical inversion; however, such sequential analytical inversions are limited to specific linear and simple cases.

**Ensemble methods**

Ensemble methods are commonly used to tackle high-dimensional problems and to approximately characterize the optimal solution. In ensemble methods, such as Ensemble Kalman filters (EnKFs) or smoothers (e.g., Whitaker and Hamill, 2002; Peters et al., 2005; Zupanski et al., 2007; Zupanski, 2005; Feng et al., 2009; Chatterjee et al., 2012), the issue of high dimensions in the system of Eq. (3) is avoided using the two following main procedures:

– observations are first assimilated sequentially in the system to reduce the dimension of the observation space, making it possible to explicitly compute matrix products and inverses, and thus propagating information from the target space to the observation space; the overall inversion period is processed incrementally using a smaller running assimilation window including a manageable number of observations; intermediate inversions are solved on the smaller running window that is gradually moved

from the beginning to the end of the overall data assimilation window; the running assimilation window with so-called analysis and forecast steps introduces complex technical challenges to rigorously propagate errors from one iteration of the running window to the next one; moreover, the sequential assimilation of observations is valid only under the assumption that observations errors are not cor-related between assimilation windows, which may prove incorrect for high-density data sets, but is an assumption also done in, e.g., variational inversions. For very dense observations, such as datasets from new-generation high-resolution satellites, the sequential assimilation of observations may not be sufficient, or at least methods may be needed to make the observation errors between sequential assim-ilation windows independent, for example by applying a whitening transformation to the observations to form a new set with uncorrelated errors as suggested by Tippett et al. (2003).The challenge is exac-erbated for long-lived species such as $CO_2$, for which assimilation windows must be long enough to maintain the propagation of information on the fluxes on long distances through transport; propagating a covariance matrix from assimilation windows to assimilation windows as accurate as possible could in principle limit the later issue, as suggested in Kang et al. (2011, 2012), but could still prove hard to apply in very high resolution problems.

 – the posterior distribution at a given step of the filter is then characterized explicitly by applying Eq. (1) on each member of the ensemble; the new intermediate posterior distribution is then sampled and propagated to the next data assimilation window.

In the atmospheric inversion community, another ensemble method is widely used, based on the Carbon-Tracker system (Peters et al., 2005), which uses an Ensemble Square Root Filter (EnSRF; Whitaker and Hamill, 2002). In that approach, the observations are split using running data assimilation windows as for other ensemble methods, but instead of directly characterizing the posterior distribution from the ensemble, the statistics of the ensemble is used to solve the analytical equation, Eq. (3), approximately. Thus, the En-SRF method is less general as EnKFs methods, as it relies on the Gaussian assumption, as well as limited non-linearity in the inversion problem, but proves very efficient at computing an approximated solution of the inversion problem. Matrix products in Eq. (3) involving the target vector covariance matrix $\mathbf{B}$ ($\mathbf{HBH}^\mathrm{T}$ and $\mathbf{BH}^\mathrm{T}$) are approximated by reducing the space of uncertainties to a low-rank representation; this is done in practice by using a Monte Carlo ensemble of possible target vectors sampling the distribution $\mathcal{N}(\mathbf{x}^\mathrm{b}, \mathbf{B})$; with such an approximation, matrix products can be written as follows:

$$\left\{ \begin{array}{rl} \mathbf{HBH}^\mathrm{T} & \simeq \dfrac{1}{N-1} \left( \mathcal{H}(\mathbf{x}_1), \mathcal{H}(\mathbf{x}_2), \ldots, \mathcal{H}(\mathbf{x}_N) \right) . \left( \mathcal{H}(\mathbf{x}_1), \mathcal{H}(\mathbf{x}_2), \ldots, \mathcal{H}(\mathbf{x}_N) \right)^\mathrm{T} \\ \mathbf{BH}^\mathrm{T} & \simeq \dfrac{1}{N-1} \left( \mathbf{x}_1, \mathbf{x}_2, \ldots, \mathbf{x}_N \right) . \left( \mathcal{H}(\mathbf{x}_1), \mathcal{H}(\mathbf{x}_2), \ldots, \mathcal{H}(\mathbf{x}_N) \right)^\mathrm{T} \end{array} \right. \tag{5}$$

where $N$ is the size of the ensemble.

From there, Eq. (1) is solved analytically by replacing $\mathbf{HBH}^\mathrm{T}$ and $\mathbf{BH}^\mathrm{T}$ by their respective approxima-tions.

By using random sampling, ensemble methods are able to approximate large dimensional matrices at a reduced cost without using the adjoint of the observation operator (see variational inversion below) that can be challenging to implement. Small ensembles generally cause the posterior ensemble to collapse, i.e., the posterior distribution is dominated by one or a very small number of members, which does not allow for a reliable assessment of the posterior uncertainties (Morzfeld et al., 2017); moreover, small ensembles introduce spuriousness in the posterior uncertainties, with irrealistic correlations being artificially generated.. In the EnSRF, small ensembles rather cause a misrepresentation of uncertainty structures, which limits the accuracy of the computed solution, and may require fixes as described in, e.g., Bocquet (2011). In any case, the level of approximation necessary for this approach to work is strongly different for each problem, which requires preliminary studies before consistent application. In particular, the so-called localization of the ensemble can be used to improve the consistency of the inversion outputs (e.g., Zupanski et al., 2007; Babenhauserheide et al., 2015).

In the current version, only the EnSRF approach is implemented in the CIF. One should note that the EnSRF, as a direct approximation of the analytical solution, can be very sensitive to non-linearity in the observation operator (e.g., Tolk et al., 2011). It can generally cope only with slight non-linearity over the assimilation window, thus, the assimilation window length has to be chosen appropriately, contrary to other ensemble methods which are usually not sensitive to non-linearity.

**Variational inversions**

Variational inversions use the fact that finding the mode of the posterior Gaussian distribution $p^{\mathrm{a}}(\mathbf{x}) \sim \mathcal{N}(\mathbf{x}^{\mathrm{a}}, \mathbf{A})$ in Eq. (2) is equivalent to finding the minimum $\mathbf{x}^{\mathrm{a}}$ of the cost function $J$:

$$J(\mathbf{x}) = \frac{1}{2}(\mathbf{x} - \mathbf{x}^{\mathrm{b}})^{\mathrm{T}}\mathbf{B}^{-1}(\mathbf{x} - \mathbf{x}^{\mathrm{b}}) + \frac{1}{2}(\mathbf{y}^{\mathrm{o}} - \mathcal{H}(\mathbf{x}))^{\mathrm{T}}\mathbf{R}^{-1}(\mathbf{y}^{\mathrm{o}} - \mathcal{H}(\mathbf{x})) \tag{6}$$

In variational inversions, the minimum of the cost function in Eq. (6) is numerically estimated iteratively using quasi-Newtonian algorithms based on the gradient of the cost function:

$$\nabla J_{\mathbf{x}} = \mathbf{B}^{-1}.(\mathbf{x} - \mathbf{x}^{\mathrm{b}}) + \mathcal{H}^{*}\left(\mathbf{R}^{-1}.(\mathbf{y}^{\mathrm{o}} - \mathcal{H}(\mathbf{x}))\right) \tag{7}$$

Quasi-Newtonian methods are a group of algorithms designed to compute the minimum of a function iteratively. It should be noted that in high-dimension problems, it can take a very large number of iterations to reach the minimum of the cost function $J$, forcing the user to stop the algorithm before convergence, thus reaching only an approximation of $\mathbf{x}^{\mathrm{a}}$; in addition, iterative algorithms can reach local minima without ever reaching the global minimum, making it essential to thoroughly verify variational inversion results; this can happen in non-linear cases, but also, due to numerical artefacts, in linear cases (some points in the cost function have gradients so close to zero that the algorithm sees them as convergence points, whereas the unique

global minimum is somewhere else). In the community, examples of quasi-Newtonian algorithms commonly used are the Broyden–Fletcher–Goldfarb–Shanno (BFGS) algorithm (Zheng et al., 2018; Bousserez et al., 2015), M1QN3 (Gilbert and Lemaréchal, 1989), and the CONGRAD algorithm (applicable only to linear or linearized problems; Fisher, 1998; Chevallier et al., 2005) based on the Lanczos method, which iter-
ates to find the eigenvalues and eigenvectors of the Hessian matrix, which is then used (in a single step) to calculate the analysis vector, $\mathbf{x}^a$. In general, quasi-Newtonian methods require an initial regularization, or "pre-conditioning" of $\mathbf{x}$, the vector to be optimized, for better efficiency. In atmospheric inversions, such a regularization is generally made by optimizing $\chi = \mathbf{B}^{-1/2}.(\mathbf{x} - \mathbf{x}^b)$ instead of $\mathbf{x}$; we note $\mathfrak{A}$ the regulariza-
tion space: $\chi \in \mathfrak{A}$. This transformation translates in Eq. (6) and (7) as follows:

$$
\begin{cases}
J_\chi &= \frac{1}{2}\chi^T\chi + \frac{1}{2}(\mathbf{y}^o - \mathcal{H}(\mathbf{B}^{1/2}.\chi + \mathbf{x}^b))^T\mathbf{R}^{-1}(\mathbf{y}^o - \mathcal{H}(\mathbf{B}^{1/2}.\chi + \mathbf{x}^b)) \\
\nabla J_\chi &= \chi + \mathbf{B}^{1/2}.\mathcal{H}^*\left(\mathbf{R}^{-1}.(\mathbf{y}^o - \mathcal{H}(\mathbf{B}^{1/2}.\chi + \mathbf{x}^b))\right)
\end{cases}
\tag{8}
$$

Solving Eq. (6) and (7) in the target vector space or Eq. (8) in the regularization space is mathematically fully equivalent, but the solution in the regularization space is often reached in fewer iterations. Moreover, in the regularization space, one can force the algorithm to focus on the main modes of the target vector space by filtering the smallest eigenvalues of the matrix $\mathbf{B}$. This reduces the dimension of $\chi$ and accelerates further the rate of convergence, although the solution of the reduced problem is only an approximation of the solution of the full problem. In the following we thus prefer calling the "regularization space" the "reduction space". The link between the two can be written as follows:

$$
\begin{aligned}
\chi_{\text{full}} &= \mathbf{Q}\mathbf{\Lambda}^{-1/2} \quad (\mathbf{x} - \mathbf{x}^b) \\
\chi_{\text{reduced}} &= \mathbf{Q}'\mathbf{\Lambda}'^{-1/2} \quad (\mathbf{x} - \mathbf{x}^b)
\end{aligned}
\tag{9}
$$

with $\mathbf{B}^{1/2} = \mathbf{Q}\mathbf{\Lambda}^{1/2}\mathbf{Q}^T$, $\mathbf{Q}$ and $\mathbf{\Lambda}$ being the matrices of the eigenvector and the matrix of the correspond-
ing eigenvalues of the matrix $\mathbf{B}$. $\mathbf{Q}'$ and $\mathbf{\Lambda}'$ are the reduced matrices of eigenvalues and eigenvectors with a given number of dominant eigenvalues.

Overall, variational inversions are a numerical approximation to the solution of the inversion problem: they involve the gradient of the cost function in Eq. (7) and require to run forward and adjoint simulations iteratively (e.g., Meirink et al., 2008; Bergamaschi et al., 2010; Houweling et al., 2016, 2014; Fortems-
Cheiney et al., 2021; Chevallier et al., 2010, 2005; Thompson and Stohl, 2014; Monteil and Scholze, 2019; Wang et al., 2019).

The variational formulation does not require calculation of complex matrix products and inversions, con-
trary to the analytical inversion, and is thus not limited by vector dimensions. Still, the inverses of the uncertainty matrices $\mathbf{B}$ and $\mathbf{R}$ need to be computed, potentially prohibiting the use of very large and/or com-

plex general matrices; this challenge is often overcome by reducing $\mathbf{B}$ and $\mathbf{R}$ to manageable combinations of simple matrices (e.g., Kronecker products of simple shape covariance matrices; see Sect. 2.3.1).

When the observation operator is linear, the posterior uncertainty matrix $\mathbf{A}$ is equal to the inverse of the Hessian matrix at the minimum of the cost function. In most cases the Hessian cannot be computed

explicitly, because of memory limitations, which is a major drawback of variational inversions. But some variational algorithms such as CONGRAD provide a coarse approximation of the Hessian: in the case of CONGRAD based on the Lanczos method, leading eigenvectors of the Hessian can be computed, together with their eigenvalues (Fisher, 1998). The approximation of the posterior uncertainty matrix $\mathbf{A}$ in the case of CONGRAD reads as follows:

$$\mathbf{A} = Hess(J)^{-1}_{\mathbf{x}^{\mathrm{a}}} \approx \mathbf{V}^{\mathrm{T}}_{\mathbf{x}^{\mathrm{a}}} \mathbf{\Lambda}^{-1}_{\mathbf{x}^{\mathrm{a}}} \mathbf{V}_{\mathbf{x}^{\mathrm{a}}} \tag{10}$$

with $\mathbf{V}_{\mathbf{x}^{\mathrm{a}}}$ the dominant eigenvectors of the Hessian matrix at the point $\mathbf{x}^{\mathrm{a}}$ and $\mathbf{\Lambda}_{\mathbf{x}^{\mathrm{a}}}$ the matrix of the dominant eigenvalues of the Hessian matrix. Please note that the dominant eigenvalues of the Hessian matrix correspond to components with low posterior uncertainties in $\mathbf{A}$.

Another approach to quantify the posterior uncertainty matrix $\mathbf{A}$, valid for both linear and non-linear

cases, is to carry out a Monte Carlo ensemble of independent inversions with sampled prior vectors from the prior distribution $\mathcal{N}(\mathbf{x}^{\mathrm{b}}, \mathbf{B})$ (e.g., Liu et al., 2017). An ensemble of posterior vectors are inferred and used to compute the posterior matrix as follows:

$$\mathbf{A} \approx \frac{1}{N-1} (\mathbf{x}^{\mathrm{a}}_1 - \mathbf{x}^{\mathrm{a}}_{ref},\ \mathbf{x}^{\mathrm{a}}_2 - \mathbf{x}^{\mathrm{a}}_{ref},\ \dots\ \mathbf{x}^{\mathrm{a}}_N - \mathbf{x}^{\mathrm{a}}_{\mathrm{ref}}) \cdot (\mathbf{x}^{\mathrm{a}}_1 - \mathbf{x}^{\mathrm{a}}_{ref},\ \mathbf{x}^{\mathrm{a}}_2 - \mathbf{x}^{\mathrm{a}}_{ref},\ \dots\ \mathbf{x}^{\mathrm{a}}_N - \mathbf{x}^{T}) \tag{11}$$

with $N$ the size of the Monte Carlo ensemble, $\mathbf{x}^{\mathrm{a}}_i$ the posterior vector corresponding to the prior $\mathbf{x}^{\mathrm{b}}_i$ of the

20 Monte Carlo ensemble and $\mathbf{x}^{\mathrm{a}}_{ref}$ the average over sampled posterior vectors.

### 2.2.2 Auxiliary computation modes

**Forward simulations**

Forward simulations simply use the observation operator to compute simulated observation equivalents. It reads as:

$$(\mathbf{x}^{\mathrm{b}},\ \mathbf{y}^{\mathrm{o}}) \rightarrow \mathcal{H}(\mathbf{x}^{\mathrm{b}}) \tag{12}$$

This mode is used to make quick comparisons between observations and simulations to check for inconsistencies before running a full inversion. It is also used by the analytical inversion mode to build response functions.

**Test of the adjoint**

The test of the adjoint is a crucial diagnostic for any inversion system making use of the adjoint of the observation operator. Such a test is typically required after making any edits to the code (to the forward observation operator or its adjoint) before running an inversion. Coding an adjoint is prone to errors and even small errors can have significant impacts on the computation of the gradient of the cost function in Eq. (7). Thus, one needs to make sure that the adjoint rigorously corresponds to the forward. This test consists in checking the definition of the mathematical adjoint of the observation operator. It writes as follows for a given target vector $\mathbf{x}$ and incremental target perturbation $\delta\mathbf{x}$:

$$< d\mathcal{H}_{\mathbf{x}}(\delta\mathbf{x}) \,|\, d\mathcal{H}_{\mathbf{x}}(\delta\mathbf{x}) > = < \delta\mathbf{x} \,|\, (\mathcal{H}^* \circ d\mathcal{H}_{\mathbf{x}})(\delta\mathbf{x}) > \tag{13}$$

$d\mathcal{H}_{\mathbf{x}}(\delta\mathbf{x})$ is the linearization of the observation operator $\mathcal{H}$ at the point $\mathbf{x}$ for a given increment $\delta\mathbf{x}$; it is computed with the tangent linear model, which is the numerical adaptation of $d\mathcal{H}_{\mathbf{x}}(\delta\mathbf{x})$. Then, $(\mathcal{H}^* \circ d\mathcal{H}_{\mathbf{x}})(\delta\mathbf{x})$ is calculated with the adjoint of the tangent-linear of $\mathcal{H}$ at the point $\mathbf{x}$.

In practice, the two terms of the equation are rarely exactly equal. Nevertheless, the difference should never exceed a few times the machine epsilon. Besides, Eq. (13) should be verified for any given target vector and increment. In practice, it is not possible to explicitly verify all possible combinations; but as the result of the test is highly sensitive to any error in the code, it is assumed that a few typical couples $(\mathbf{x}, \delta\mathbf{x})$ are sufficient to certify the validity of the adjoint.

## 2.3 Identification of common elementary transformations

### 2.3.1 General purpose operations

Each inversion algorithm and computation mode mentioned above can be decomposed into a pipeline of elementary transformations. These transformations are listed in Tab. 1 and include: the observation operator and its adjoint (their matrix representations in linear cases), matrix products with target and observation error covariance matrices and corresponding adjoints, and random sampling of normal distributions. To limit redundancy in the CIF as much as possible, these elementary transformations are included in the CIF as generic transformation blocks on the same conceptual level. Overall, the decomposition of computation modes presently implemented in the CIF into elementary transformations leads to the structure in Fig. 1.

Avoiding redundancy makes the maintenance of the code much easier, and provides a clear framework for extensions to other inversion methods or features. For instance, inverse methods based on probability density functions other than normal distributions could be easily implemented by updating the random ensemble generator, or by implementing new cost functions representing non-Gaussian distributions, while keeping the remaining code unmodified. In particular, non-Gaussian cost functions still rely on the compu-

tation of the observation operator. New combinations of elementary transformations can also directly lead to new methods. For instance, ensemble variational inversion (e.g., Bousserez and Henze, 2018) is a direct combination of the available variational pipeline and the random sampling pipeline. Inversions estimating hyper-parameters through maximum-likelihood or hierarchical Bayesian techniques (e.g., Michalak et al., 2005; Berchet et al., 2014; Ganesan et al., 2014) could be integrated in the CIF by adapting the Gaussian cost function and by implementing a corresponding computation pipeline.

The complexity of the selected elementary transformations spans a wide range, from one-line straightforward codes to computationally expensive and complex code implementation. In small dimensional and/or linear problems, the computation of the observation operator using its Jacobian and matrix products may be computationally expensive, but is in principle rather straightforward to implement. For non-linear and/or high-dimensional problems, these transformations require simplifications and numerous intermediate steps. For instance, applying matrix products to the error covariance matrix $\mathbf{R}$ and $\mathbf{B}$ and computing their inverse is easy in small dimensions, but can be limiting in high dimensional problems. For that reason, the error covariance matrices are often reduced to particular decompositions; for instance, the error covariance matrix on the target vector $\mathbf{B}$ is often written as a Kronecker product of multiple spatial and/or temporal covariance matrices of lower dimensions, making matrix products manageable (e.g., Chevallier et al., 2005; Meirink et al., 2008; Yadav and Michalak, 2013).

In any case, the observation operator (see details in Sect. 2.3.2) appears as the center piece of any inversion method.

## 2.3.2 Observation operator

The observation operator is a key component of all inversion methods. It links the target space to the observation space, and conversely, its adjoint links the observation space to the target space. To do so, the observation operator projects its inputs through various intermediate spaces to the outputs. As atmospheric inversions need a representation of the atmospheric transport (and chemistry if relevant) to link the target vector (including surface fluxes, atmospheric sources and sinks, initial and boundary conditions for limited domains and time-windows, etc.) to the observation vector (including some form of atmospheric concentration measurements), the observation operator is built around a given CTM in most cases: Eq. (14) illustrates the various projections in the common case.

$$
\mathbf{x} \xrightarrow{\mathbf{\Pi}_{\mathcal{X}}^{\widetilde{\mathfrak{F}}}} \mathbf{f} \xrightarrow{\mathbf{\Pi}_{\widetilde{\mathfrak{F}}}^{\mathcal{F}}} \text{inputs} \xrightarrow{\text{model}} \text{outputs} \xrightarrow{\mathbf{\Pi}_{\mathcal{C}}^{\mathfrak{M}}} \mathbf{c} \xrightarrow{\mathbf{\Pi}_{\mathfrak{M}}^{\mathcal{Y}}} \mathcal{H}(\mathbf{x}) \tag{14}
$$

with $\mathbf{f}$ the target vector projected at the CTM's resolution (includes fluxes, but also other types of inputs required by the CTM), $\mathbf{c}$ the raw outputs extracted from the run of the CTM's executable (in general 4-dimensional concentration fields). $\mathbf{\Pi}$ operators are intermediate projectors: $\mathbf{\Pi}_{\mathcal{X}}^{\widetilde{\mathfrak{F}}}$ projects the target vector at

the spatial and temporal resolutions of the CTM's inputs, $\Pi_{\mathfrak{F}}^{\mathcal{F}}$ dumps the input vector in files usable by the CTM's executable, $\Pi_{\mathcal{C}}^{\mathfrak{M}}$ reads the CTM's outputs, $\Pi_{\mathfrak{M}}^{\mathcal{Y}}$ reprojects the raw outputs at the observation vector resolution (mostly the temporal resolution as the model and the observation worlds do not follow the same time line).

The targeted structure of the CIF should allow a full flexibility of observation operators, from the straightforward widely-used decomposition detailed in Eq. (14) to more elaborated approaches including multiple transport models and/or complex super-observations (e.g., in Bréon et al., 2015; Staufer et al., 2016, authors implemented differences between observation sites and time in the observation vector instead of observations from individual sites in order to focus on spatial/temporal gradients, thus allowing to limit the influence of
background concentrations in the computation of local fluxes) and hyper-parameters (e.g., emission factors and model parameters used to produce emission maps; Rayner et al., 2010; Asefi-Najafabady et al., 2014). Therefore, the observation operator is designed as a pipeline of elementary interchangeable transformations with standardized input and output formats such that:

$$\mathcal{H} = \mathcal{H}_1 \circ \mathcal{H}_2 \circ \cdots \circ \mathcal{H}_N \tag{15}$$

In such a formalism, all intermediate transformations have the same conceptual level in the code. They are functions ranging from spatial reprojection, to temporal interpolations, to more complex operations such as the reconstruction of satellite total columns from concentrations simulated at individual levels in the transport model. In the CIF, all these transformations have the same input and output structure and, thus, their order can be changed seamlessly to execute a given configuration. Please note that the commutative
property of elementary transformations as pieces of code does not guarantee the commutative property of the corresponding physical operators.

Such a transformation-based design allows us to rigorously separate transformations and thus to implement and test their respective adjoints more easily. Once adjoints for each individual operation are implemented, the construction of the general adjoint is straightforward by reversing the order of forward opera-
tions:

$$\mathcal{H}^* = \mathcal{H}_N^* \circ \mathcal{H}_{N-1}^* \circ \cdots \circ \mathcal{H}_1^* \tag{16}$$

Fig. 2 shows an example of a typically targeted observation operator. Operators from Eq. (14) are reported for the illustration. It includes two numerical models chained with each other; they can be for instance a coarse global CTM and a finer resolution regional CTM, such as in Rödenbeck et al. (2009) or Belikov et al.
(2016). The system applies a series of transformations to the target vector, including spatial deaggregation for the optimization of emissions by regions, sector deaggregation to separate different activity sectors, reprojection to the CTM's resolution (a simple interpolation of mass-conserving regridding is available so far, with

regular and irregular domains), application of temporal profiles (which is critical in air quality and anthropogenic $CO_2$ applications), unit conversions to the required inputs for the CTMs. On the observation vector side, observations can span multiple model time-steps, requiring posterior temporal averages, etc. In the case of super-observations (satellites retrievals, images, spatial gradients, etc.) in the observation vector, it is often necessary to combine multiple simulated point observations in given grid cells and time stamps into a single super-observation, to limit redundant observations, hence the size of the observation vector, but also to limit representativeness issues. Super-observations are commonly used in the case for satellite observations being compared to all the model levels above a given location; concentration gradients comparing observations at different time and locations (see e.g., Bréon et al., 2015; Staufer et al., 2016) are another example of observation aggregation to reduce representativeness errors; isotopic ratios are also super-observations as they require to simulate separate isotopologues and recombine them after the simulation (as done in e.g., van der Velde et al., 2018; Peters et al., 2018). The case of Fig. 2 also include background concentrations in the target vector: a background is often used to fix some biases in initial and lateral concentrations in limited-area models, and in observations (mostly satellites); the background variables are processed at the very end of the pipe when re-constructing the observations vector.

The mathematical formalism of Eq. (15) and (16) suggests that transformations are necessarily computed in a serialized way, thus limiting applications to simple target variables upstream the transport model. However, each elementary transformation handles components of the inputs it is concerned with, leaving the rest identical and forwarding it to later transformations. Typically, it does not actually limit applications to simple target variables upstream the CTM. For instance, in the case of target variables optimizing biases in the observations, the corresponding components of the target vector $\mathbf{x}$ are forwarded unchanged by all transformations in Fig. 2 until the very last operation, where they are used for the final comparison to the observation vector.

## 3 Practical implementation

### 3.1 General rules

The *Community Inversion Framework* project follows the organisation scheme of Fig. 3. A centralized website is available at community-inversion.eu. The website includes all information given in the present paper, as well as further documentation details, practical installation instructions, tutorials and examples of possible set-ups. To foster the collaborative dynamics of our project, all scripts and codes are available in open-access on a GitLab server at git.nilu.no/VERIFY/CIF, where updates are published regularly. The frozen version of the code, documentation and data used for the present publication is available in Berchet et al. (2020). The repository includes the documentation, sources for the CTMs implemented in the CIF, as well as the Python library *pyCIF*. Our project is distributed as an open-source project under the CeCILL-C licence of

the French law (cecill.info). The license grants full rights for the users to use, modify and redistribute the original version of the CIF, conditional to the obligation to make their modifications available to the community and to properly acknowledge the original authors of the code. The authors of modifications own intellectual property of their modifications, but under the same governing open license. Software that may

be built around the CIF in the future can have different licensing, but all parts of the code originating from the CIF will be governed by the original CeCILL-C license, hence must remain open source. Similarly, some constituting pieces of the CIF can be adapted from other softwares governed by other licenses and simply interfaced to the CIF (e.g., transport models, minimizing algorithms, etc.); in that case, the corresponding softwares keep their original license and their use and distribution in the CIF is subject to authorization by

their owners (although open distribution and integration in the standard version of the CIF is encouraged). This is the case of the CONGRAD and M1QN3 algorithms which are used as minimizing algorithms in the variational inversions of the demonstration case in Sect. 4. The M1QN3 algorithm is distributed under the GNU General Public License, whereas CONGRAD is owned by ECMWF and is not open source; the later was interfaced with the CIF but is not openly distributed.

The *pyCIF* library, written in Python 3, is the practical embodiment of the CIF project. All theoretical operations described in Sect. 2 are computed by this module. It includes inversion computations, pre- and post-processing of CTM inputs and outputs, as well as target and observation vector reprojections, aggregation, etc., as written in Eq. (15). Python coding standards follow the community standards PEP-8 (python.org/dev/peps/pep-0008/).

Test cases (including the ones presented in Sect. 4) are distributed alongside the CIF codes and scripts. To foster portability and dissemination, a dedicated Docker image is distributed with *pyCIF*, providing a stable environment to run the system and enabling full reproducibility of the results from one machine to the other.

## 3.2   Plugin-based implementation

To reflect the theoretical flexibility required in the computation of various inversion methods and observation
operators, we made the choice of implementing *pyCIF* following an abstract structure with a variety of so-called Python plugins, which are dynamically constructed and inter-connected depending on the set-up.

### 3.2.1   Objects and classes in *pyCIF*

General classes of objects emerge from the definition of the abstract structure of the inversion framework. These classes are defined by the data and metadata they carry, as well as by the methods they include and
their interaction with other classes. The main classes are the following:

- computation modes: forward computations, the test of the adjoint, variational inversions, EnSRF and analytical inversions are available (see details in Sect. 2.2);

- models: interfaces to CTMs; includes generation of input files, executing the code and post-processing outputs; included are a Gaussian model described in Sect. 4 for the demonstration of the system, as well as CHIMERE, LMDZ, FLEXPART, TM5, and STILT, all of which will be described in a dedicated future publication;

- platforms: deal with specific configurations on different clusters; it includes a standard platform as well as two supercomputers where the CIF was tested;

- target vectors: store and apply operations related to the target vector, including spatial and temporal aggregation, deaggregation, regularization of the target vector;

- observation vectors: store and apply operations related to the observation vector, including application of observation errors;

- observation operators: drive CTMs and apply elementary operations between the control and observation vectors;

- transformations: elementary operations used to build the observation operator; includes temporal averaging or deaggregating of the target and observation vectors, projection of the target vector at the model input resolution, etc.;

- data vectors: store all information on inputs for *pyCIF*; this vector is used by the observation and target vector classes to build themselves;

- minimizers: algorithms used to minimize cost functions, including M1QN3 and CONGRAD algorithms so far;

- simulators: cost functions to minimize in variational inversions; only includes the standard Gaussian cost function so far;

- domains: store information about the CTM's grid, including coordinates of grid cell centers and corners, vertical levels, etc.;

- fluxes, fields, meteo-data: fetch, read and write different formats of inputs for CTMs (surface fluxes, 3D fields and meteorological fields respectively); so far includes only inputs specific to included CTMs, but will ultimately include standard data streams, such as widely used emission inventories or meteorological fields such as those from ECMWF;

- measurements: fetch, read and write different types of observation data streams; only include the World Data Center for Greenhouse Gases so far (https://gaw.kishou.go.jp/), but classical data providers such as ICOS (icos-cp.eu) or ObsPack (Masarie et al., 2014) will also be implemented in the CIF;

satellite products, in particular the formatting of averaging kernels and other metadata, should also be included in the CIF in the near future as they play a growing role in the community.

Details on metadata and operations for each class are given in Supplements, Tab. S1. Our objective was to design a code that is fully recursive in the sense that modifying some instance of a class does not require to update other classes calling or being called by the modified class. Thus, each class is built so that it only needs internal data, as well as data from the execution level just before and after it, in order to avoid complex dependencies while allowing proper recursive behaviour in building the transformation pipe. For instance, the observation operator applies a pipe of transformations from the target vector to the observation vector. Some transformations will use the model class to run the model, or the domain class to carry out reprojections, or the target vector to aggregate/deaggregate target dimensions, etc. Despite the many complex transformations carried out under the umbrella of the observation operator, only the sub-transformations of the pipe are accessible at the observation operator level, which do not have to directly carry information about e.g., the model or other classes required at sub-levels. This makes the practical code of the observation operator much simpler and as easy to read as possible.

### 3.2.2 Automatic construction of the execution pipe

To translate the principle scheme of Fig. 1, *pyCIF* is not built in a sequential rigid manner. Plugins are interconnected dynamically at the initializing step of *pyCIF* depending on the chosen set-up (see Sect. 3.3 for details on the way to configure the CIF). The main strength of such a programming structure is the independence of all objects in *pyCIF*. They can be implemented separately in a clean manner. The developer only needs to specify what other objects are required to run the one being developed and *pyCIF* makes the links to the rest. It avoids unexpected impacts elsewhere in the code when modifying or implementing a feature in the system. In the following, we call this top-down relationship in the code a dependency.

For each plugin required in the configuration (primarily the computation mode), *pyCIF* initializes corresponding objects following simple rules. Following dependencies detailed in Tab. S1, for every object to initialize, *pyCIF* will fetch and initialize required plugins and attach them to the original plugin. If the required plugin is explicitly defined in the configuration, *pyCIF* will fetch this one. In some cases, some plugins can be built on default dependencies, which do not need to be defined explicitly in the configuration file. In that case, the required plugin can be retrieved using default plugin dependencies specified in the code itself and not needed in the configuration.

For instance, in the call graph in Fig. 1, "variational" (inversion) is a "computation mode" object in *pyCIF*. To execute, it requires a "minimizer" object (CONGRAD, M1QN3, etc.) that is initialized and attached to it. The minimizer requires a "simulator" object (the cost function) that itself will call functions in the "control

vector" object and the "observation operator" object. Then the "observation operator" will initialize a pipeline of transformations including running the "model", and so on and so forth.

## 3.3 Definition of configurations in the CIF

In practice, *pyCIF* is configured using a YAML configuration file (yaml.org). This file format was primarily chosen for its flexibility and intuitive implementation of hierarchical parameters. In the YAML language, key words are specified with associated values by the user. Indentations indicate sub-levels of parameters, which makes it a consistent tool with the coding language python.

To set-up a *pyCIF* computation, the user needs to define the computation mode and all related requirements in the YAML configuration file. Every plugin has mandatory and optional arguments. The absence of one mandatory argument raises an error at initialization. Optional arguments are replaced by corresponding default values if not specified. Examples of YAML configuration files used to carry out the demonstration cases are given in Supplement Section S3.

## 4 Demonstration case

In the following we describe a demonstration case based on a simple implementation of a Gaussian plume dispersion model and simple inversion set-ups. The purpose of this demonstration case is a proof-of-concept of the CIF, with various inversion methods. We comment and compare inversion set-ups and methods for the purpose of the exercise, but conclusions are not made to be generalized to any inversion case study due to the simplicity of our example. The test application with a simple Gaussian plume model allows users to quickly carry out the test cases themselves, even on desktop computers, to familiarize themselves with the system. Nevertheless, the Gaussian plume model is not only relevant for teaching purposes, but also for real applications, as it is used in many inversion studies from the scale of industrial sites with in-situ fixed or mobile measurements (e.g., Kumar et al., 2020; Foster-Wittig et al., 2015; Ars et al., 2017) to the larger scales with satellite measurements to optimize individual clusters of industrial or urban emissions (e.g., Nassar et al., 2017; Wang et al., 2020). Other models implemented in the CIF will be presented in a future paper evaluating the differences when using different transport models with all other elements of the configuration identical. The purpose of such an evaluation is to produce a rigorous inter-comparison exercise identifying the effect of transport errors in inversion systems.

## 4.1 Gaussian plume model

Gaussian plume models approximate real turbulent transport by a stable average Gaussian state (Hanna et al., 1982). Such models are not always suitable to compare with continuous measurements but can be adapted when using observations averaged over time. In the following, we consider the Gaussian plume assumption

to be valid for comparing to hourly averaged observations. A simple application of the Gaussian plume model was implemented in the CIF as a testing and training utility. It is computationally easy to run, even on desktop computers. It includes the most basic Gaussian plume equations. In that application, concentrations $\mathcal{C}$ at location $(x_0,\ y_0,\ z_0)$ downwind from a source of intensity $f$ at $(x_1,\ y_1,\ z_1)$ are given by:

$$5 \quad \mathcal{C}(x_0,\ y_0,\ z_0) \ = \ \frac{f}{2\,\pi\,.\,\sigma_y\,.\,\sigma_z\,.\,\bar{u}}\exp\left(-\frac{y^2}{\sigma_y^2}\right).\exp\left(-\frac{z^2}{\sigma_z^2}\right) \tag{17}$$

with

$$\begin{cases} \sigma_z & = & a\,.\,x^b \\ \sigma_y & = & |465.11628\times\ x\,.\,\tan(0.017653293\,(c-d\,.\,\ln x))| \\ x & = & <\dfrac{\mathbf{u}}{\bar{u}}\ |\ v_{\text{(source, receptor)}}\ > \\ y & = & (\dfrac{\mathbf{u}}{\bar{u}}\ \times\ v_{\text{(source, receptor)}}\ ) \end{cases} \tag{18}$$

$x$ is the downwind distance between the source and receptor points along the wind axis, $y$ is the distance between the wind axis and the receptor point; , $v_{\text{(source, receptor)}}$ is the vector linking the source and the receptor point. $z$ is the difference between the source and the receptor altitudes. $\mathbf{u}$ is the vectoral wind speed, with $\bar{u}$ is the average wind speed in the domain of simulation. $<\ \cdot\ |\ \cdot\ >$ and $(\ \cdot\times\cdot\ )$ depict the scalar and the vector products respectively. $(a,b,c,d)$ are parameters depending on the Pasquill-Gifford atmospheric vertical stability classes. There are 7 Pasquill-Gifford stability classes, from A extremely unstable (mostly in summer during the afternoon) to G very stable (occurring mostly during nighttime in winter). As the purpose of the demonstration case is primarily to work on coarsely realistic concentration fields, with a computational cost as low as possible, our implementation of the Gaussian plume model does not include any representation of particle reflection on the ground or on the top of the planetary boundary layer.

To illustrate atmospheric inversions, we use a grid of point surface fluxes to simulate concentrations using the Gaussian plume equation. Thus, the concentration at a given point and time $t$ is the sum of Gaussian plume contributions from all individual grid points:

$$\begin{aligned} \mathcal{C}(x_0,\ y_0,\ z_0,t) \ &= \ \sum_{(x_1,\ y_1,\ z_1)\in\ \text{grid}} \frac{f(x_1,\ y_1,\ z_1,t)}{2\,\pi\,.\,\sigma_y(t)\,.\,\sigma_z(t)\,.\,\bar{u}(t)}\exp\left(-\frac{y^2}{\sigma_y(t)^2}\right)\exp\left(-\frac{z^2}{\sigma_z(t)^2}\right) \\ &= \ \sum_{(x_1,\ y_1,\ z_1)\in\ \text{grid}} \mathbf{H}_{(x_1,\ y_1,\ z_1,\ t)}\times f(x_1,\ y_1,\ z_1,\ t) \\ &= \ \mathbf{H}(t)\,.\,\mathbf{f}(t) \end{aligned} \tag{19}$$

This formulation highlights the linear relationship between concentrations and fluxes. As the concentrations can be expressed as a matrix product, the computation of the adjoint of the Gaussian model is straightforward and does not require extra developments:

$$
\begin{aligned}
\delta f(x_1, y_1, z_1, t) & = \sum_{(x_0, y_0, z_0) \in \text{obs}} \frac{\delta C(x_0, y_0, z_0, t)}{2\pi\,\sigma_y(t)\,\sigma_z(t)\,\bar{u}(t)} \exp\left(-\frac{y^2}{\sigma_y(t)^2}\right) \exp\left(-\frac{z^2}{\sigma_z(t)^2}\right) \\
& = \mathbf{H}(t)^T . \mathbf{C}(t)
\end{aligned}
\tag{20}
$$

For the purpose of our demonstration cases, meteorological conditions (wind speed, wind direction, and stability class) are randomly generated for the simulation time-window. Fixed seeds are selected for the generation of random conditions in order to make them reproducible.

## 4.2   Configuration

### 4.2.1   Modelling set-up

Cases discussed in Sect. 4.3 are based on the Gaussian model computed on a domain of $2.5 \times 2\,\text{km}^2$ with a grid of $18 \times 12$ grid cells. Surface point sources are located at the center of corresponding grid cells, with flux intensities as represented in Fig. 4. Five virtual measurement sites are randomly located in the domain with randomly selected altitudes above ground level as shown in Fig. 4. The inversion time-window spans a period of five days with hourly observations and meteorological forcing conditions. Meteorological conditions are

a combination of a wind speed, a wind direction and a stability class applicable to the whole simulation domain for each hour. The three parameters are generated randomly for the period, without respect for realistic relatively smooth transitions in weed speed and direction and stability class.

Truth observations are generated by running the Gaussian model in forward mode with known fluxes defined as the sum of prior fluxes $\mathbf{f}$ (used later in the inversions) in Eq. (21) and an arbitrary perturbation as

defined in Eq. (22), and illustrated in Fig. 4 (left and right respectively).

$$
\mathbf{f} = f_0 . \left\{ \cos\left(2\pi\,\frac{x}{\sigma_x^1}\right) + \sin\left(2\pi\,\frac{y}{\sigma_y^1}\right) + \left(\frac{x}{\sigma_x^2}\right)^2 + \left(\frac{y}{\sigma_y^2}\right)^2 \right\}
\tag{21}
$$

$$
\delta\mathbf{f} = 0.2 \times f_0 . \left\{ \cos\left(2\pi\,\frac{x}{\sigma_x^3}\right) + \sin\left(2\pi\,\frac{y}{\sigma_y^3}\right) \right\}
\tag{22}
$$

with $f_0$ an arbitrary reference flux, and scaling lengths $\sigma_x^1, \sigma_x^2, \sigma_x^3, \sigma_y^1, \sigma_y^2, \sigma_y^3$ equals 500, 1000, 200, 1000, 1000 and 300 m respectively. Reference fluxes and perturbations are constant over time.

A random noise of $1\%$ of the standard deviation of the forward simulations was added to the truth observations to generate measurements. Please note that the perturbation of the fluxes is generated using an explicit formula and not a random perturbation with a given covariance matrix. We discuss results with different possible target vectors and covariance matrices.

### 4.2.2 Inversion set-ups

The objective of our test case is to prove that our system enables users to easily compare the behaviour of different inversion methods in various configurations. To do so, we conduct three sets of four inversions for the demonstration of our system. Each set includes one analytical inversion, one EnSRF-based inversion and two variational inversions based on M1QN3 and CONGRAD minimization algorithms respectively. The sequential aspect of the EnSRF is not implemented in the CIF, hence the comparison with the other inversion methods only includes the random sampling of the target vector distribution to solve Eq. (5).

The three sets of inversions differ by the dimension of the target vector and the spatial correlations of errors. The first set uses a target vector based on a grid of $3 \times 3$ pixels-aggregated regions or "bands" independent from each other i.e. with no spatial error correlations. The target vectors of the second and third sets are defined at the grid cell's resolution with horizontal isotropic error correlations, following an exponential decay with a horizontal scale of $500\,\text{m}$ and $200000\,\text{m}$ respectively; the latter case is used to demonstrate that the implementation of correlation lengths is correct as very long correlations are equivalent to having only one spatial scaling factor in the target vector. For all inversion set-ups, the target vectors are defined as constant over time, i.e., only one coefficient per spatial dimension is optimized for the 5 days $\times$ 24 hours, computed by the model. In all set-ups, the magnitude of the observation noise used to generate "true" observations is chosen as observation errors in the matrix $\mathbf{R}$ for consistency. In all cases, the diagonal terms of the $\mathbf{B}$ matrix are set to 100%.

To assess the sensitivity of each set-up to the allocated computational resources, we computed the EnSRF and the two variational inversions with varying numbers of simulations $N$. In the case of the EnSRF, $N$ simply depicts the size of the Monte Carlo ensemble. For variational inversions, each step i.e each computation of the cost function and its gradient requires one forward simulation and one adjoint simulation. The Gaussian model is a simple auto-adjoint model, which makes the adjoint simulations as long as the forward one. Therefore, $N$ is equal to twice the number of computations of the cost function (one for the forward and one for the adjoint) in the minimization algorithm. Note that in many real application cases, the adjoint of a CTM is more costly than the forward, reducing the number of iterations possible in $N$ times the cost of a forward. Indeed, despite the adjoint being mathematically as expensive as the forward, in practice, the computation of adjoint operations often requires the re-computation of intermediate forward computations, therefore increasing the computational burden of the adjoint model. More precisely, users and developers of adjoint transport models choose the number of forward re-computations to be carried out, based on a space-speed trade off: by saving all forward intermediate states, the adjoint is as costly as the forward, but the disk space burden can be extremely challenging to manage, thus increasing the overall computation time in return.

## 4.3 Results and discussion

In the following, we present detailed figures for the test case at the pixel resolution with a correlation length of 500 m. For the sake of readability, figures for other test cases are grouped in Sect. S2 of the Supplement.

Posterior increments are presented in Fig. 5. Observation locations and heights are reported for information. The color scale of flux increments is the same as in Fig. 4 which represent the true "increments" to be retrieved. In Fig. 8, we present the evolution of the cost function of Eq. (6) depending on the number of simulations used for each inversion method for the three demonstration cases (see details on the corresponding number of simulations of each inversion methods in Sect. 4.2.2). The x-axis has been cropped at the origin as the EnSRF value for small sizes of random ensembles diverges to infinity.

In the case with the target vector aggregated on groups of pixels, all inversion methods converge towards a very similar solution. In this case, the posterior increments reproduce the overall structure of the truth-prior difference, with one local minimum in the center of the domain. However, the aggregated target vector results in too coarse patterns which are not representative of the actual true-prior difference. In the case with the target vector at the grid's resolution with spatial correlations of 500 m, all methods capture well the true-prior difference structure. However, posterior increments are rather noisy compared to the truth. This is due to the spatial correlations being inconsistent with the smooth perturbation with fixed length scales in Eq. (22). Correlations help smoothing the posterior fluxes but not perfectly consistently with the truth. For the case with the target vector at the grid's resolution with spatial correlations of 200000 m, all methods converge towards a very smooth and similar solution, consistently with what is expected with a very long correlation length. However, they do not converge towards the same solution, probably because a larger number of iterations/members would be needed to fully converge.

In all cases, CONGRAD converges at a faster pace than the other two methods and, after a limited number of iterations, the convergence rate is close to zero, suggesting additional simulations do not provide significant additional information to CONGRAD (although additional iterations bring new constraints on the posterior uncertainty matrix).

Overall, CONGRAD appears to be the most cost-efficient algorithm to estimate the analytical solution in the case of a linear inversion in our very simple demonstration case. Though not as efficient, the EnSRF method can approximate the analytical solution at a reduced cost. By design, its computation can easily be parallelized, which can allow a faster computation than CONGRAD when computational resources are available in parallel. M1QN3 proves not as efficient as its CONGRAD counterpart, but contrary to CONGRAD, it can accommodate non-linear cases.

The reduction of uncertainties and posterior uncertainty matrices are shown in Fig. 6 and 7, and equivalents in Supplement. Regarding posterior uncertainties, CONGRAD proves relatively efficient to approximate the analytical solution, especially at the pixel resolution. The variational inversion with Monte Carlo and

M1QN3 computations and the inversion with EnSRF are much noisier. Approximating posterior matrices requires a large number of Monte Carlo members and proves very challenging in real-world applications.

## 5 Conclusions

We have introduced here a new generic inversion framework that aims at merging existing inversion systems together, in order to share development and maintenance efforts and to foster collaboration on inversion studies. It has been implemented in a way that is fully independent from the inversion configuration: from the application scales, from the species of interest, from the CTM used, from the assumptions for data assimilation, as well as from the practical operations and transformations applied to the data in pre- and post-processing stages. This framework will prevent redundant developments from participating research groups and will allow for a very diverse range of applications within the same system. New developments will be made in an efficient manner with limited risks of unexpected side effects, and thanks to the generic structure of the code, specific developments for a given application can be directly applied to other applications. For instance, new inversion methods implemented in the CIF can be directly tested with various transport models. With modern inversion methods moving towards an hybrid paradigm of variational and ensemble methods, the flexibility of the CIF will be a valuable asset as abstract methods can be easily mixed with each other.

We have presented the first version of this *Community Inversion Framework* (CIF) alongside with its python-dedicated library *pyCIF*. As a first step, analytical inversions, variational inversions with M1QN3 and CONGRAD, and EnSRF have been implemented to demonstrate the general applicability of the CIF. The four inversion techniques were tested here on a test case with a Gaussian plume model and with observations generated from known "true" fluxes. The impact of spatial correlations and of spatial aggregation, which drive the shape of the control vectors used in this scientific community, has been illustrated. The analytical inversion is the most accurate approach to retrieve the true fluxes, as expected, followed by variational inversions with the CONGRAD algorithm in our simple linear case. In our simple case, EnSRF and M1QN3 generally take longer to converge towards the true pattern of the fluxes, even though EnSRF inversions have the advantage to be fully parallelizable, in contrast to variational inversions, that are sequential by design and therefore harder to parallelize (e.g., Chevallier, 2013).

The next step of the CIF is the implementation of a large variety of CTMs. The implementation of new CTMs already interfaced with other inversion systems should not bring particular conceptual challenges as all interface operations are already written in their original inversion system; in most cases, re-arranging existing routines is sufficient to interface a model to the CIF. One particular challenge concerns I/O optimizations: the generation of inputs and the processing of outputs can be time consuming and in some very heavy applications require specific numerical and coding optimizations. The very general formalism of the

CIF may hamper the ability of applying these particular optimizations for some models. Best efforts will have to be deployed to take full advantage of advanced I/O and data manipulation libraries in python to limit this weakness.

CHIMERE, LMDz, TM5, FLEXPART, and STILT have already been implemented and a sequel paper will evaluate and compare their behaviour in similar inversion set-ups. COSMO-GHG and WRF-CHEM are also planned to be implemented in the near future to widen the developer/user community of the system. The use of various CTMs in identical inversion configurations (inversion method, observation and target vector, consistent interface, etc.) will allow extensive assessments of transport errors in inversions. Despite many past efforts put in inter-comparison exercises, such a level of inter-comparability has never been reached and will be a natural by-product of the CIF in the future. In addition, comparing posterior uncertainties from different inversion methods and set-ups will make it possible to fully assess the consistency of different inversion results.

The flexibility of the CIF allows very complex operations to be included easily. They include the use of satellite observations, that will be evaluated in a future paper, inversions using observations of isotopic ratios and optimizing both surface fluxes and source signatures (Thanwerdas et al., 2021). In addition, even though greenhouse gas studies have been the main motivation behind the development of the CIF, the system will also be tested for multi-species inversions including air pollutants.

*Code and data availability.* The codes, documentation pages (including installation instructions and tutorials) and demonstration data used in the present paper are registered under the following DOI: 10.5281/zenodo.4322372 (Berchet et al., 2020).

*Author contributions.* All authors contributed to the elaboration of the concept of Community Inversion Framework. Antoine Berchet designed the structure of the CIF. Antoine Berchet, Espen Sollum, Isabelle Pison and Joël Thanwerdas are the main developers of the python library pyCIF. Antoine Berchet, Espen Sollum and Isabelle Pison maintain the documentation and websites associated to the Community Inversion Framework.

*Acknowledgements.* The Community Inversion Framework is currently funded by the project VERIFY (verify.lsce.ipsl. fr), which received funding from the European Union's Horizon 2020 research and innovation programme under grant agreement no. 776810.

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

**Table 1.** Elementary operations required for each data assimilation method. An = Analytical inversion; EnKF = Ensemble Kalman filter; Var = Variational; Fwd = Forward simulation; AdTest = Test of the adjoint. We note $\mathcal{X}$ and $\mathcal{Y}$ the target and observation spaces respectively, $\mathfrak{A}$ the regularization space in the minimization algorithm of variational inversions; the $(\cdot)^*$ symbol depicts the adjoint of corresponding spaces.

| Elementary operation | Mathematical formulation | An | EnKF | Var | Fwd | AdTest |
|---|---|---|---|---|---|---|
| Forward observation operator | $\begin{aligned} \mathcal{X} &\to \mathcal{Y} \\ \mathbf{x} &\to \mathcal{H}(\mathbf{x}) \text{ or } \mathbf{Hx} \end{aligned}$ | X | X | X | X | X |
| Adjoint observation operator | $\begin{aligned} \mathcal{Y}^* &\to \mathcal{X}^* \\ \delta\mathbf{y} &\to \mathcal{H}^*(\delta\mathbf{y}) \text{ or } \mathbf{H}^T \delta y \end{aligned}$ | X | | X | | X |
| Normalisation of the observation increments | $\begin{aligned} \mathcal{Y}^* &\to \mathcal{Y}^* \\ \delta\mathbf{y} &\to \mathbf{R}^{-1}\delta\mathbf{y} \end{aligned}$ | | | X | | |
| Regularization of the target space | $\begin{aligned} \mathfrak{A} &\to \mathcal{X} \\ \chi &\to \mathbf{x} = \mathbf{B}^{1/2}\chi + \mathbf{x}^{\mathrm{b}} \end{aligned}$ | | | X | | |
| Adjoint of the target space regularization | $\begin{aligned} \mathcal{X}^* &\to \mathfrak{A}^* \\ \delta\mathbf{x} &\to \delta\chi \equiv \mathbf{B}^{1/2}\delta\mathbf{x} \end{aligned}$ | | | X | | |
| Target space sampling | $\begin{aligned} \mathcal{X} \times \mathcal{X}^2 &\to \mathcal{X}^N \\ (\mathbf{x}, \mathbf{B}) &\to (\mathbf{x}_1, \mathbf{x}_2, \ldots, \mathbf{x}_N) \end{aligned}$ | | X | | | |

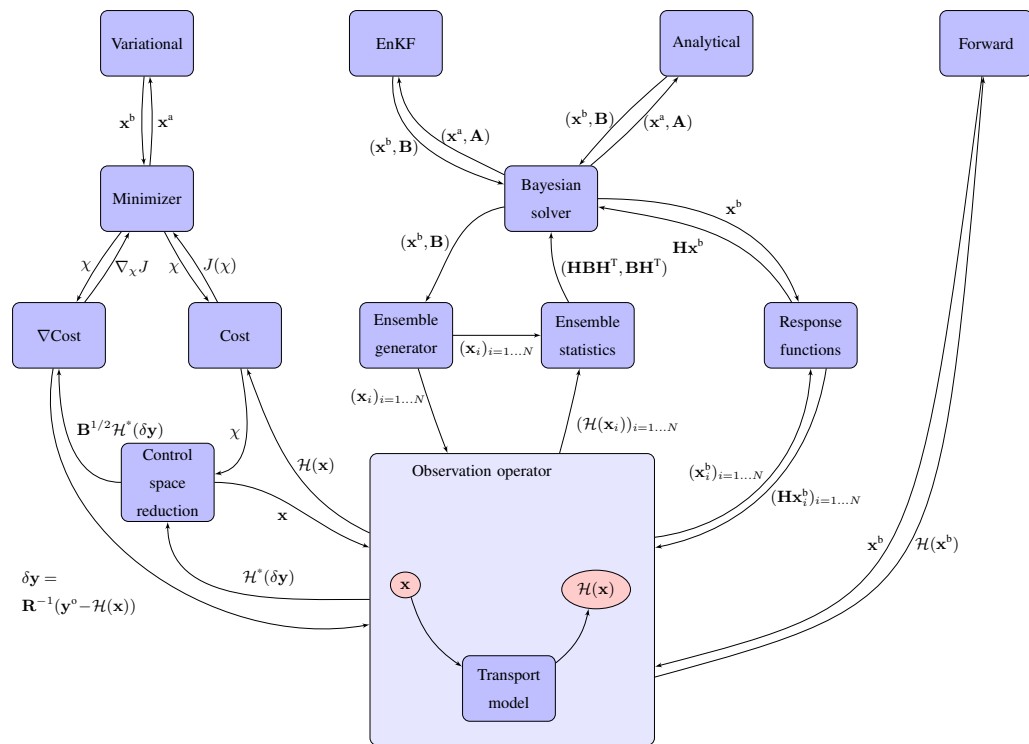

**Figure 1.** Call structure of the CIF.

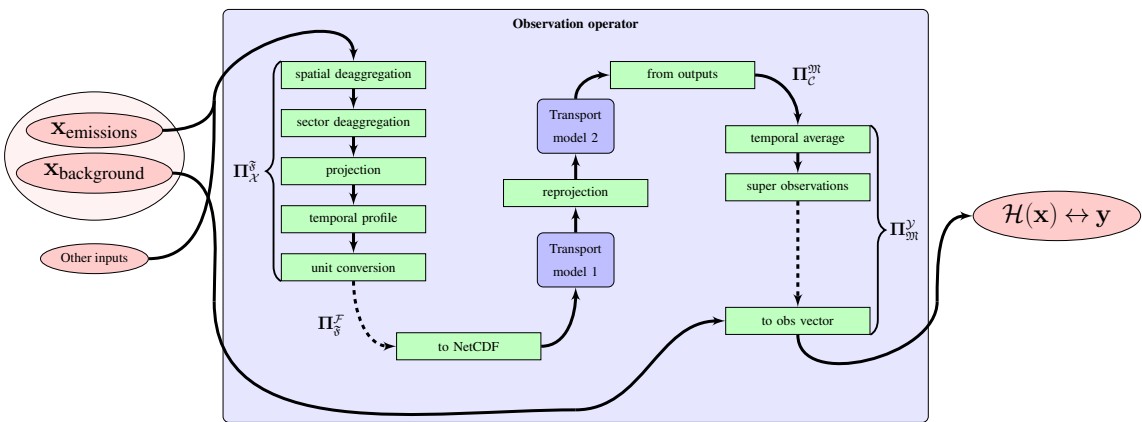

**Figure 2.** Observation operator structure. Emissions are processed from the target vector to generate model inputs, as well as other inputs, no optimized by the inversion; in this example, some background for the simulations is also optimized by the inversion and is added to simulations at the end of the pipeline just before stacking outputs to the observation vector format.

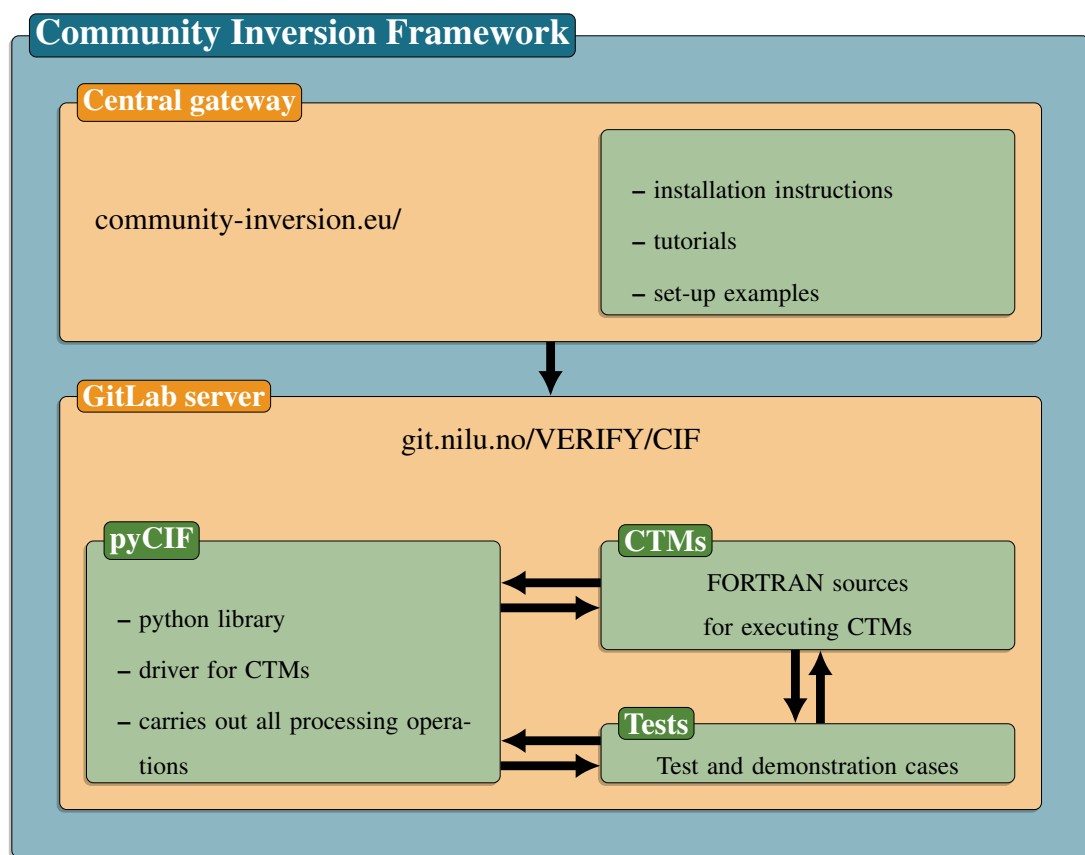

**Figure 3.** Organisation of the Community Inversion Framework.

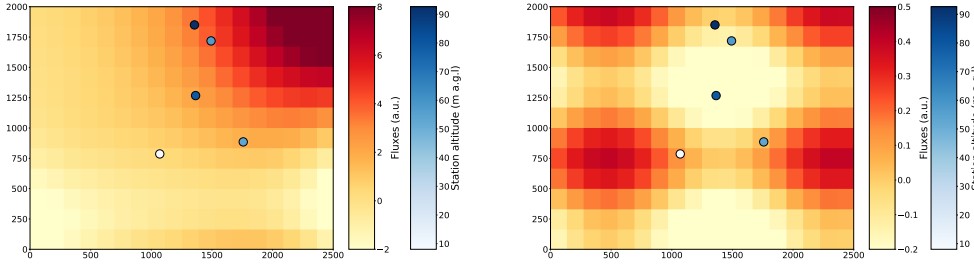

**Figure 4.** (left) Prior fluxes and observation sites. (right) Perturbation from the prior used to generate "true" observations. Observation sites are shown as circles coloured according to their height in meters above ground level (m a.g.l.). Fluxes are reported in arbitrary units (a.u.)

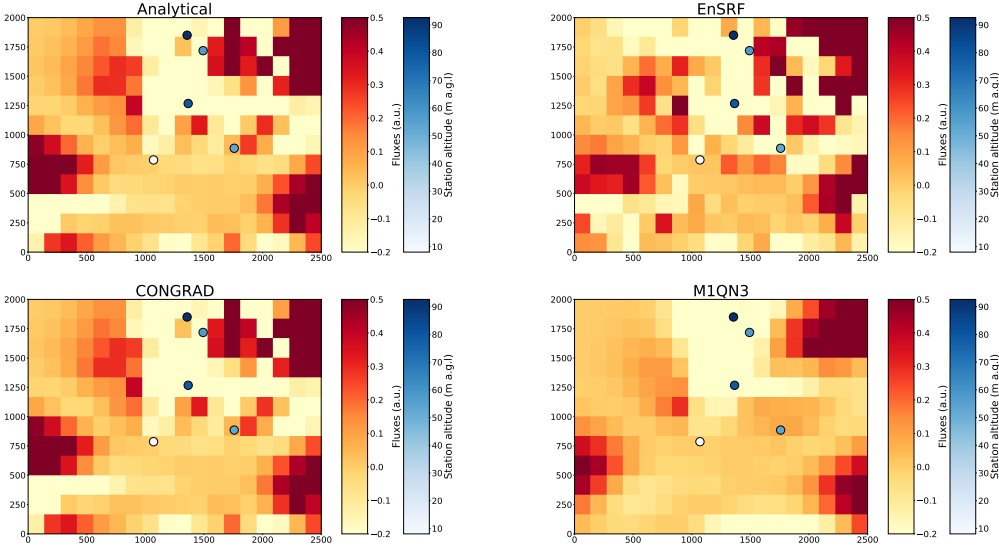

**Figure 5.** Posterior increments for analytical, EnSRF, variational with CONGRAD and variational with M1QN3 (from top to bottom, left to right) for an inversion set-up at the pixel resolution with horizontal correlation length of 500 m.

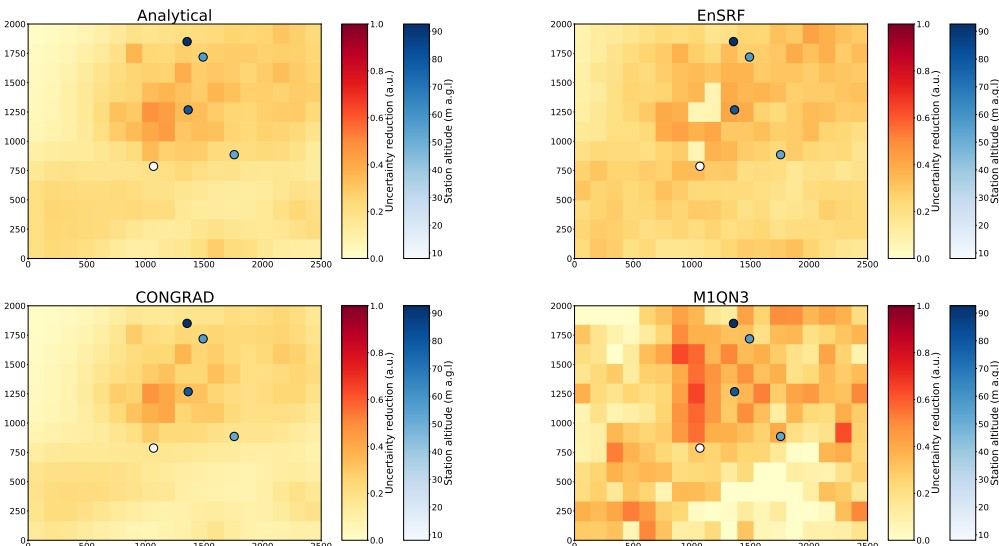

**Figure 6.** Uncertainty reduction for analytical, EnSRF, variational with CONGRAD and variational with M1QN3 (from top to bottom, left to right) with the same set-up as in Fig. 5.

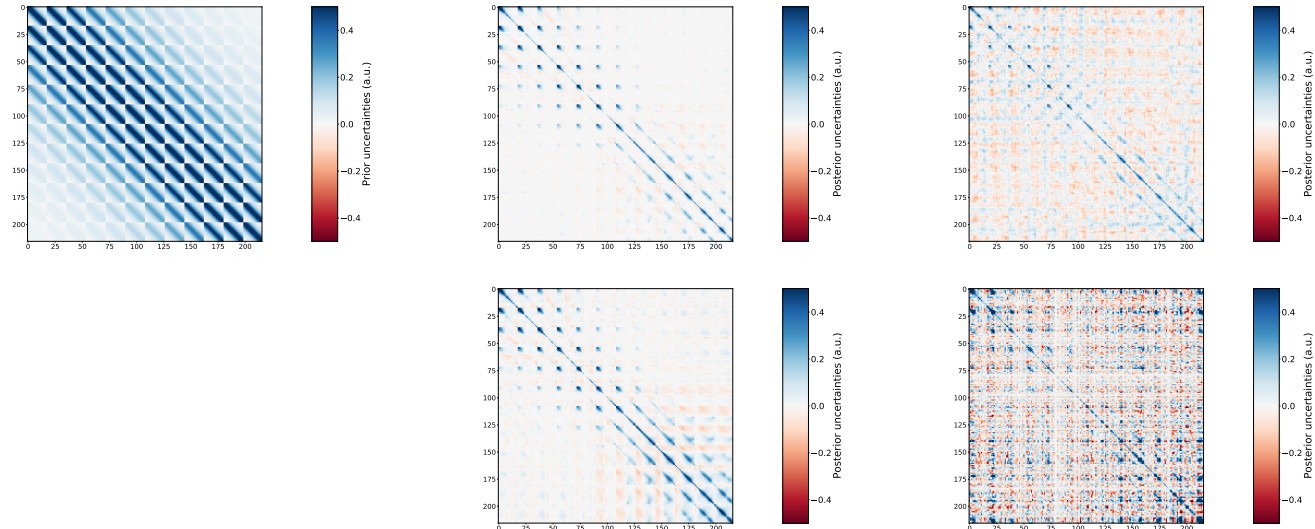

**Figure 7.** Prior (left) and posterior (middle and right) uncertainty matrices for analytical, EnSRF, variational with CON-GRAD and variational with M1QN3 (from top to bottom, middle and right columns), with the same set-up as in Fig. 5. All matrices are reported with unitless values, i.e., a "1" on the diagonal corresponds to a 100% uncertainty.

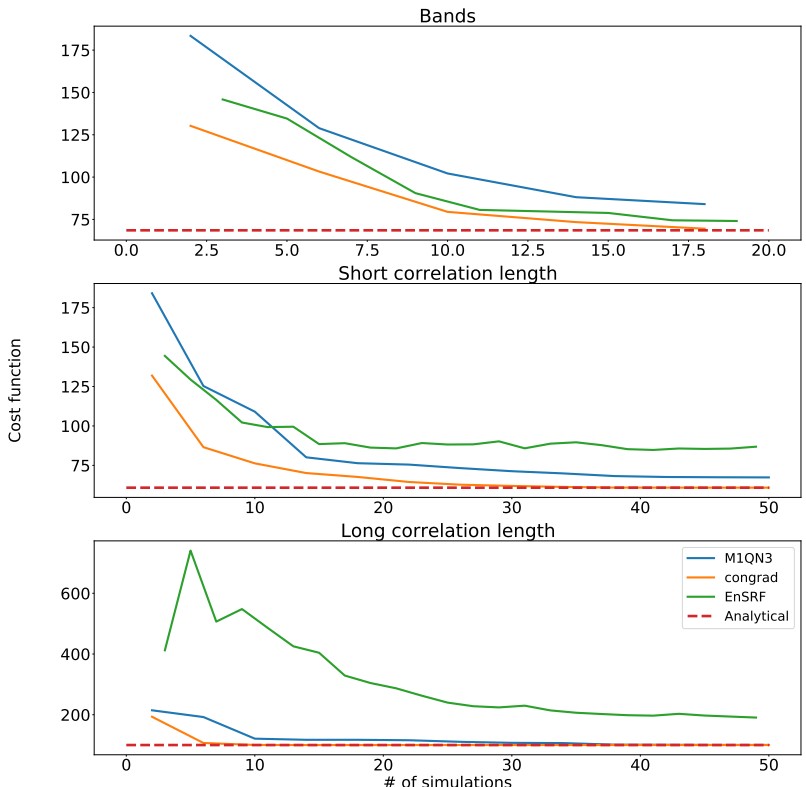

**Figure 8.** Cost function evaluation for varying numbers of computed simulations for analytical (red), EnSRF (green), variational with CONGRAD (orange) and variational with M1QN3 (blue) methods. (top) inversion set-up with aggregated regions of 3 pixels × 3 pixels ; (middle) inversion set-up at the pixel resolution with horizontal correlation length of 500 m; (bottom) inversion set-up at the pixel resolution with horizontal correlation length of 200000 m.