# Peer review of "The Community Inversion Framework v1.0: a unified system for atmospheric inversion studies"

_Geoscientific Model Development, 2020_

## Referee Comment (RC1) · Anonymous Referee #1 · 17 Feb 2021

The submitted paper presents the Community Inversion Framework (CIF) to help rationalise development efforts and leverage the strengths of individual inversion systems into a comprehensive framework. The CIF is primarily a programming protocol to allow various inversion bricks to be exchanged among researchers. The system is supposed to allow running different atmospheric transport models, different observation streams and different data assimilation approaches. The paper describes a system that will bring a major software advances for future inversion frameworks. The presentation is clear and well written. I have few main major concerns detailed in the main comments below.

[Figure]

Main comments:

The paper advertises and the multi-CTM or model capability, but this is not shown in the paper only mentioning a future paper. This capability needs to be showcased in the paper or such claims should be removed or rephrased.

It is a bit limiting to narrow the scope only on GHG as inversion of reactive species and primary aerosols is quite relevant as well. Especially if CIF plans to, or already includes the CHIMERE CTM which is originally designed for air quality applications.

The ensemble and analytical results point out possible problems in the implementation of the EnKF and in the analytical solution using correlation length scales. It is not entirely clear on how the system uses the ensemble information to perform the inversion. This ensemble-based inversion should not be called an EnKF as it is not sequential. This needs to be revised or diagnosed or explained.

Finally, the testing of system using the Gaussian plumes on my local machine (Mac) was impossible for the time I gave in. I regularly use python and successfully setup conda on my local machine. But it seems that the CIF is dependent on many specific packages that bring conflict issues and are not straightforward to install (specifically GDAL seems problematic). It required some research to find different commands to install the packages from what is indicated on the online documentation. After trying for hours, I had to give up as my time to revise this paper is limited.

Specific comments:

P3, L8: Define better what a unified system would be here.

P3, L12: OOPS main goals are on NWP. The CAMS system will run on OOPS (which is based on IFS) which will run inversions in the future. Maybe it is worth mentioning the CoCO2 projects (formerly CHE).

The authors should also mention the Joint Effort for Data Integration led by UCAR/JCSDA. https://www.jcsda.org/jcsda-project-jedi.

P3, L13-14: DART is an EnKF (EAKF to be more precise) that allows to run on the CESM earth system model. There is also a WRF mode. I would not call this unified or modular system compared to OOPS, CIF or JEDI.

Also, DART can run atmospheric composition and recently chemical species inversions as well (see Gaubert et al., 2020).

P5, L10: Some of the models mentioned here use semi-lagrangian advection scheme. Maybe remove Eulerian and use a different word.

P5, L11: Mizzi et al., 2016 is a WRF CHEM paper and also do not perform any source inversion but only concentration assimilation. Please consider removing this reference.

Equation 2: Does N indicates a normal distribution? Please clarify. Also, in the second brace element, is this zero in the first moment? If yes, I guess this assumes unbiased observation and prior. I know this is a requirement for data assimilation formulation. But often not the case in practice in atmospheric composition. Maybe you can detail and develop this in the text.

P6, L22-25: This is an important point, maybe the authors could expand more on this for the non-data assimilation expert?

P7, L8: Are control space and target space the same thing or different? Please be consistent (I would recommend control space as it is more generally used in atmospheric DA) or explain the differences.

P7, L12: This part of the sentence is not very clear. Replace "... problems such as the variational and the ensemble Kalman filter methods which are described below."

P7, L18: "limited non-linearity": This is not necessarily true, or it is misleading. EnKFs are frequently applied successfully on chemical transport models which can be significantly non-linear systems in the polluted boundary layer for example.

P7, L25: "running computation window": I believe the authors refer to what is commonly

called assimilation window? (Note that inversion is part of data assimilation techniques, so it is fine to call it data assimilation window in the inversion framework). Please be consistent in the terminology in this paragraph. "Smaller", smaller than what? I understand that for GHG inversions especially CO2 you need long window given the observation network and the CO2 atmospheric lifetime, but this is not general to all atmospheric composition inversion.

P7, L29-31: "for very dense ... may not be sufficient": This is not necessarily true, or the statement is misleading. This also needs to be proven. If the reference exists please add it. One can foresee that with a better coverage and higher satellite sensitivity towards the surface the window length could be reduced hence the number of observations to go through sequentially is reduced. Please correct or remove this statement or justify this more clearly.

P8, L11: "under-estimating": This is not necessarily true. Small ensemble size can over and underestimate the uncertainty. More importantly they introduce spuriousness. Please clarify the statement.

P8, L13-14: Could you possibly provide few examples of different problems having different approximations?

P8 L15-28: I recommend revising this introduction of the variational method. Add a sentence to present the cost function. Put the equation right after. Explain in few sentences that the aim is to minimize this cost function and that to achieve this the gradient is calculated. Then put the gradient formula.

P8, L25: I am not sure this is true. If the state vector is large, i.e. increasing resolution or augmenting the control variables, or increasing the number of observations, the minimization is expected to be slower.

P12, Equation 12: Please, be explicit about what the index represents. "... of elementary interchangeable N transformations ..."

P13, L10: It is a bit of pity that air quality application is only mentioned here. I think a system such as CIF could greatly benefit the air quality community as much as the GHG community, which are getting more connected lately.

P13, L13: I am not sure what the authors meant by spatial gradient as observations? Please clarify or remove.

P13, L14-15: Maybe the authors could mention here that the main idea behind "super-obing" is to lower the discrepancies of the representativeness between model and observations.

P13, L15-16: "This is the case for . . . time and locations," This sentence is very unclear. Please rephrase or explain better.

P15, L21-31: What about satellite observations? I think the authors should add statement about this.

P17, L11: Those studies use Gaussian plume technique but do not provide a continuous inversion constrain globally. Please correct the statement as it makes the reader believes that the Gaussian plume technique would allow a global inversion similar to a CTM based inversion.

P19, L28: 10000m is 4 times the size of the domain here. No data assimilation system would use such long length scale. Is this on purpose? If yes, please justify.

P20, L16, Fig 5 and similar other figures. With all the fluxes and fluxes differences plots it would be better to use python colour maps that are diverging and centred around 0. This will improve the readability of the figures. P20, L20: I am unsure if I understand correctly but I believe the authors are speaking about iterations in the variational sense and members in the EnKF sense. If yes, please have few sentences to explain what the word 'simulation' means. Alternatively, I would replace the word 'simulations' by 'iterations or members.'

P20, L32: 10000m is 4 times the size of the domain here. See the related com-

ment above. Such long length scales would provide very uniform structure on the increments. Structure similar to the 500m increments is seen on the result using the 10000m length scale. I expect there is typo through the text a 0 should be removed to 1000m?

P20,L33-P21,L2: This look to me more like a bug rather noise as vertical lines or "chess board" like pattern appear in the increments results. Such pattern also occurs in the analytical solution. This not looking like sampling noise to me. I think this points out possible problems in the implementation of the EnKF and of the systems other than the variational methods. It is not entirely clear if the system uses the ensemble information to derive jacobians for the calculation of H and perform a minimization? In this case this inversion cannot be called EnKF. This needs to be revised or diagnosed or explained.

Technical comments:

P2, L27-29: an order in all those references, chronological or alphabetical.

P7, L16: replace "... linear, simple cases." by "... linear and simple cases."

P7, L19: change "characterized" to "characterize".

P9, L13: Chevallier et al., 2005: put this reference to the previous parenthesis for clarity.

P10, L5: There is a colon here but nothing after. Should we expect equations?

P15, L18: I understand what the authors means but this need to be reformulated. "Meteos" is not really correct English.

P19, L3-10: Maybe add the equations in the text where they are mentioned.

P22, L6: This is not a fair statement considering on how the "EnKF" has been implemented here. It doesn't seem to be strictly an EnKF as the sequential assimilation is not performed. Consider changing or removing this in the conclusion. Also, this is not the scope of GMD paper here to compare methods but to describe and showcase

model/software developments for geoscience.

P22, L10-12: The authors could add some sentences about the challenges in implementing actual CTMs in the CIF framework. For example, among many other challenges, I/O is a limiting factor for data assimilation especially with CTMs.

References:

Gaubert, B., Emmons, L. K., Raeder, K., Tilmes, S., Miyazaki, K., Arellano Jr., A. F., Elguindi, N., Granier, C., Tang, W., Barré, J., Worden, H. M., Buchholz, R. R., Edwards, D. P., Franke, P., Anderson, J. L., Saunois, M., Schroeder, J., Woo, J.-H., Simpson, I. J., Blake, D. R., Meinardi, S., Wennberg, P. O., Crounse, J., Teng, A., Kim, M., Dickerson, R. R., He, H., Ren, X., Pusede, S. E., and Diskin, G. S.: Correcting model biases of CO in East Asia: impact on oxidant distributions during KORUS-AQ, Atmos. Chem. Phys., 20, 14617–14647, https://doi.org/10.5194/acp-20-14617-2020, 2020.

---

## Referee Comment (RC2) · Peter Rayner (Referee) · 22 Feb 2021

article times natbib amsmath

**General Comments**

This paper introduces a software system to facilitate ensembles of inverse calculations. The likely focus is atmospheric inversions in which model inputs such as surface fluxes are inferred from combinations of prior information and observations. Such a system

is clearly useful since ensembles based on choices of ingredients (such as models or input covariances) provide a better basis for both the means and uncertainties of posterior estimates. the paper is certainly within scope for GMD since it describes a system which might well be widely used in the future. It is also well written though some final proofreading, especially around pluralisation, will be needed.

I note I am not reviewing the software itself. There is only one major critique of the paper. I think the benefits of the system are slightly over-sold (but what software re-lease doesn't do this). I believe, for example. that CIF doesn't provide a framework for comparing estimates where the target spaces differ such as ENKF with short windows vs long-term mean approaches. That's an intellectual/mathematical task which one could instantiate in software. this would probably involve classes representing PDFs which might be internally represented by ensembles or parameters and functions then defining transformations among these PDFs. That quibble aside there is no doubt the system will help such comparisons.

On the positive side the ability to configure the computational pipeline is a marvellous innovation. Building our own, less ambitious system, we have struggled to make it agnostic about computation.

Perhaps more important than supporting intercomparisons, the tools provided as part of the system will lower the entry barrier for new groups entering the field. This was a surprising spin-off from the TRANSCOM effort 20 years ago. Most systems since then have been too closely tied to a given model for universal use so CIF can provide a major boost to the field. The fact that it is well documented will only help this.

There is one aspect, though, that seems seriously deficient. If I read correctly, methods for calculating uncertainty in the Gaussian posterior are not provided. One important role of a new framework like CIF is to educate, to encourage good practice. That includes calculating and engaging with the uncertainty in the results. Certainly any of the analytic methods under-estimate this term but it is still true that under-sampling of

concentration is a major source of uncertainty for inversions at all scales. I request that the authors at least detail a roadmap for including posterior uncertainty.

**Specific Comments**

**P7 bottom** this correlation of errors is a common critique of ensemble methods but I've rarely seen it implemented in a non-ensemble method either and it's perfectly possible to do in ensemble methods, one just adds some extra state variables carrying observational corrections and build the correct autocorrelations into them instead

**P8L22** saying "the maximum probability" and "the mode" is tautological I think

**P9** congratulations for explicitly writing out the regularisation/reduction step which is so often assumed. It is called pre-conditioning so often that you should add the term for clarity

**P9** it's worth commenting that leading eigen-vectors of the Hessian correspond to the low uncertainty parts of the solution which might or might not be what you want

**P11** citing the Bousserez and Henze paper is a good advertisement for your approach, you might want to say in the abstract or intro that many new methods are hybrid and this combined framework simplifies implementing such methods

**P15** a technical question, how do you handle target variables which aren't needed at various stages, e.g. bias corrections in observations which aren't inputs to the CTM? the pipelining structure doesn't lend itself well to this problem.

**P18** I think Eq. 17, do you consider sources as points or area integrals, if so do you need to integrate over the area of the source, especially for nearby observations?

**P20** it's worth adding that the amount of intermediate calculation needed is at user discretion; a space-speed trade-off. CMAQ, for example needs almost none, having checkpointed what it needs on the forward run it had to do anyway at the start of the iteration.

**References**

Cressie, N. and Johannesson, G.: Fixed rank kriging for very large spatial data sets, Journal of the Royal Statistical Society: Series B (Statistical Methodology), 70, 209–226, 2008.
Palmer, P. I., Suntharalingam, P., Jones, D., Jacob, D. J., Streets, D. G., Fu, Q., Vay, S. A., and Sachse, G. W.: Using CO2: CO correlations to improve inverse analyses of carbon fluxes, Journal of Geophysical Research: Atmospheres, 111, doi:10.1029/2005JD006697, 2006.
Rayner, P. J., Michalak, A. M., and Chevallier, F.: Fundamentals of data assimilation applied to biogeochemistry, Atmospheric Chemistry and Physics, 19, 13 911–13 932, doi:10.5194/acp-19-13911-2019, https://www.atmos-chem-phys.net/19/13911/2019/, 2019.

---

## Author Comment (AC1) · 30 Jun 2021

**Author responses to reviews to "The Community Inversion Framework v1.0: a unified system for atmospheric inversion studies"**

Antoine Berchet[1,*], Espen Sollum[2], Rona L. Thompson[2], Isabelle Pison[1], Joël Thanwerdas[1], Grégoire Broquet[1], Frédéric Chevallier[1], Tuula Aalto[3], Adrien Berchet[14], Peter Bergamaschi[4], Dominik Brunner[5], Richard Engelen[6], Audrey Fortems-Cheiney[1], Christoph Gerbig[7], Christine D. Groot Zwaaftink[2], Jean-Matthieu Haussaire[5], Stephan Henne[5], Sander Houweling[8], Ute Kartens[9], Werner L. Kutsch[10], Ingrid T. Luijkx[11], Guillaume Monteil[9], Paul I. Palmer[12], Jacob C. A. van Peet[8], Wouter Peters[11,13], Philippe Peylin[1], Elise Potier[1], Christian Rödenbeck[7], Marielle Saunois[1], Marko Scholze[9], Aki Tsuruta[3] and Yuanhong Zhao[1]

[1]Laboratoire des Sciences du Climat et de l'Environnement, CEA-CNRS-UVSQ, Gif-sur-Yvette, France
[2]Norwegian Institute for Air Research (NILU), Kjeller, Norway
[3]Finnish Meteorological Institute (FMI), Helsinki, Finland
[4]European Commission Joint Research Centre, Ispra (Va), Italy
[5]Swiss Federal Laboratories for Materials Science and Technology (Empa), Dübendorf, Switzerland
[6]European Centre for Medium-Range Weather Forecasts, Reading, RG2 9AX, UK
[7]Max Planck Institute for Biogeochemistry, Jena, Germany
[8]Vrije Universiteit Amsterdam, Department of Earth Sciences, Earth and Climate Cluster, Amsterdam, the Netherlands
[9]Dep. of Physical Geography and Ecosystem Science, Lund University, Sweden
[10]Integrated Carbon Observation System (ICOS-ERIC), Helsinki, Finland
[11]Meteorology and Air Quality Group, Wageningen University and Research, Wageningen, the Netherlands
[12]School of GeoSciences, University of Edinburgh, Edinburgh, EH9 3FF, UK
[13]Centre for Isotope Research, University of Groningen, Groningen, the Netherlands
[14]Institut Pprime (UPR 3346, CNRS-Université de Poitiers-ENSMA), Bat H2, 11 Boulevard Marie et Pierre Curie, TSA 51124, 86073, Poitiers Cedex 9, France

**Correspondence:** antoine.berchet@lsce.ipsl.fr

We thank the referees for their detailed and fruitful comments on our manuscript. We reproduce below their reviews and embed our responses in bold blue text in their comments. Revised section of the manuscript are reproduced in italic blue. We also provide a track-change manuscript at the end of the present document.

**1  Referee #1**

5  The submitted paper presents the Community Inversion Framework (CIF) to help rationalise development efforts and leverage the strengths of individual inversion systems into a comprehensive framework. The CIF is primarily a programming protocol to allow various inversion bricks to be exchanged among researchers. The system is supposed to allow running different atmospheric transport models, different observation streams and different data assimilation approaches. The paper describes a

system that will bring a major software advances for future inversion frameworks. The presentation is clear and well written. I have few main major concerns detailed in the main comments below.

**1.1 Main comments**

The paper advertises and the multi-CTM or model capability, but this is not shown in the paper only mentioning a future paper.
5  This capability needs to be showcased in the paper or such claims should be removed or rephrased.

**The CIF is a multi-purpose systems. The two main purposes are to enable the comparisons of different inversion methods with a given inversion case, and the comparison of different transport models for a given inversion configuration. We decided to split the two aspects into two different papers for clarity. We clarified the text accordingly**

*p4. l. 14: In the present paper, we lay out the basis of the CIF, giving details on its underlying principles and overall*
10  *implementation. The proof-of-concept focuses on the implementation of several inversion methods, illustrated with a test case. We will dedicate a future paper to the evaluation of the system on a real-life problem with a number of interfaced atmospheric (chemistry-)transport models.*

It is a bit limiting to narrow the scope only on GHG as inversion of reactive species and primary aerosols is quite relevant as well. Especially if CIF plans to, or already includes the CHIMERE CTM which is originally designed for air quality
15  applications.

**The CIF is primarily designed by members of the GHG community, thus the bias in presentation. However all the conceptual environment of the CIF can be applied directly to chemistry, as it is the case in CHIMERE, which solves complex chemical schemes reproducing the full chemistry of O3, NO2, VOC, CO, etc., as well as in LMDZ e.g. with simple chemical schemes around methane, CO or nitrous oxide. We clarified the text accordingly**

20  *p3 l34: In particular, although primarily designed for GHGs applications, the CIF is based on a general structure that will allow applications to air quality data assimilation.*

*Air quality applications are also mentioned throughout the introduction more explicitly.*

The ensemble and analytical results point out possible problems in the implementation of the EnKF and in the analytical solution using correlation length scales.

25  **There was indeed a difference in the way the correlations were computed in the analytical and the ensemble method, versus the variational method. We fixed the issue and also updated the test cases to better highlight the consistency of the results.**

It is not entirely clear on how the system uses the ensemble information to perform the inversion. This ensemble-based inversion should not be called an EnKF as it is not sequential. This needs to be revised or diagnosed or explained.

30  **We agree that in this application there is no time-stepping in the EnKF solution and hence the word Filter is not appropriate. However, the implementation of the solution methods allows it, and algorithmically follows the Ensemble Square Root Filter originally described in Whitaker and Hamil, (2002) and later used in Peters et al., (2005). The important distinction from the other methods is the use of the statistics of the ensemble to solve the Kalman equations,**

**and the possibility to solve non-linear problems. We think the name EnSRF is therefore suited, and will use it in the revised version. We also include the note about the time-stepping.**

*The section on Ensemble methods (p8-10) has been extensively rewritten to clarify the difference between EnSRF and other ensemble methods. See track-change manuscript*

5  Finally, the testing of system using the Gaussian plumes on my local machine (Mac) was impossible for the time I gave in. I regularly use python and successfully setup conda on my local machine. But it seems that the CIF is dependent on many specific packages that bring conflict issues and are not straightforward to install (specifically GDAL seems problematic). It required some research to find different commands to install the packages from what is indicated on the online documentation. After trying for hours, I had to give up as my time to revise this paper is limited.

10  **We are sorry to hear about difficulties to install our system. We will give more details on the installation in the documentation web pages. To make the testing of our system easier, we have now set up a docker image (publicly available here: https://hub.docker.com/repository/docker/pycif/pycif-ubuntu) with all packages correctly installed and demonstration data available as well. Thus, new users can simply download the docker image and run it on their system, avoiding complicated configuration of libraries.**

15  **The docker image is provided with the corresponding configuration command lines to be run on a standard Ubuntu system to obtain the same environment. These lines can be used to adapt to other OS and distributions.**

**More specifically, we agree that the GDAL library can create some issues when not correctly installed. However, GDAL is standard in geoinformation systems and such a library is essential for all geographical operations, such as reprojecting from one grid to another, or finding in which grid cell an observation point is. In very regular grids, such**

20  **a library is not necessary, but the CIF is design to accommodate many transport models and input data, which are not necessarily always on regular lon/lat grid.**

*The docker image is now mentioned in Sect 3.1 p.17, l19: To foster portability and dissemination, a dedicated Docker image is distributed with pyCIF, providing a stable environment to run the system and enabling full reproducibility of the results from one machine to the other.*

25  ## 1.2  Specific comments

– P3, L8: Define better what a unified system would be here.

*Rephrased p3 l11 as: A unified system, as a community platform running multiple transport models, with diverse inversion methods, would provide ...*

– P3, L12: OOPS main goals are on NWP. The CAMS system will run on OOPS (which is based on IFS) which will run

30  inversions in the future. Maybe it is worth mentioning the CoCO2 projects (formerly CHE). The authors should also mention the Joint Effort for Data Integration led by UCAR/JCSDA. https://www.jcsda.org/jcsda-project-jedi.

**The CAMS operational system will indeed run with OOPS, to be adapted for flux inversion purposes. The CIF is part of the CoCO2 project and is used as an interface between the operational community of CAMS and the**

**research community. The CIF will also be used for the design of operational systems at the national and regional scales in Europe. We clarified this aspect and also mention the JEDI effort as follows:**

*p3 l15: "Collaborative efforts towards unified systems are already happening in other data assimilation communities, with, e.g., the Object-Oriented Prediction System (OOPS; coordinated by the European Centre for Medium-range Weather Forecast, UK), or the Joint Effort for Data Integration (led by UCAR/JCSDA; https://www.jcsda.org/jcsda-project-jedi). The Data Assimilation Research Testbed (DART) is also an example of collective effort proposing common data assimilation scripts for diverse applications (e.g., Earth system, chemical species inversions)"*

– P3, L13-14: DART is an EnKF (EAKF to be more precise) that allows to run on the CESM earth system model. There is also a WRF mode. I would not call this unified or modular system compared to OOPS, CIF or JEDI. Also, DART can run atmospheric composition and recently chemical species inversions as well (see Gaubert et al., 2020).

**We thank the reviewer for these insights and have updated the text accordingly (see reply above).**

– P5, L10: Some of the models mentioned here use semi-lagrangian advection scheme. Maybe remove Eulerian and use a different word.

**We rephrased as follows: p5 l19:**

*[. . . ] by numerical (chemistry-)transport models. These can be of various types e.g. global circulation models (e.g., LMDZ, Chevallier et al. 2010; TM5, Houweling et al.2014; GEOS-Chem, van der Laan-Luijkx et al. 2017; Liu et al. 2015; Palmer et al. 2019; Feng et al. 2017;NICAM, Niwa et al. 2017), regional Eulerian chemistry-transport models (e.g., CHIMERE, Broquet et al.2011; Fortems-Cheiney et al. 2019; WRF-CHEM, Zheng et al. 2018; COSMO-GHG, Kuhlmann et al. 2019;LOTOS-EUROS, Curier et al. 2012) or Lagrangian particle dispersion models (e.g., FLEXPART, Thompson25and Stohl 2014; STILT, Bagley et al. 2017; Brioude et al. 2013; Trusilova et al. 2010).*

– P5, L11: Mizzi et al., 2016 is a WRF CHEM paper and also do not perform any source inversion but only concentration assimilation. Please consider removing this reference.

**There was a mistake in our references indeed. We have now replaced Mizzi by Kuhlman et al. 2019.**

– Equation 2: Does N indicates a normal distribution? Please clarified. Also, in the second brace element, is this zero in the first moment? If yes, I guess this assumes unbiased observation and prior. I know this is a requirement for data assimilation formulation.But often not the case in practice in atmospheric composition. Maybe you can detail and develop this in the text.

**N indeed indicates a normal distribution. This is now stated in the text. We also clarified the underlying assumption of unbiased errors. This reads as follow now:**

*" is to assume prior and observation spaces to be normal distributions, noted $\mathcal{N}(\cdot, \cdot)$ below, the first argument representing the average of the distribution and the second argument the covariance matrix. In addition, when assuming that the distributions in the state vector space and the observation space are independent from each other, and that errors in*

*the observation and the state vector spaces have Gaussian, unbiased distributions, it is possible mathematically derive the Bayes theorem and to represent the distributions of Eq. (2) as follows:*

*EQUATION 2*

*with $\mathbf{B}$ and $\mathbf{A}$ the prior and posterior covariance matrix of uncertainties in the target vector, $\mathbf{x}^b$ and $\mathbf{x}^a$ the prior and posterior target vectors and $\mathbf{R}$ the covariance matrix of uncertainties in the observation vector and the observation operator."*

*The assumption that errors are unbiased, which makes it possible to write normal distributions in Eq. (1) with means $\mathbf{x}^b$, $\mathbf{0}$ and $\mathbf{x}^a$ respectively, is needed to simplify the formulation of the Bayesian problem in atmospheric inversions. Neglecting error biases have known impacts on inversion results (Masarie et al., 2011); they can be accounted for online as an unknown to be solved by the inversion (Zammit-Mangion et al., 2021), but are often treated offline from the inversion, either through pre-processing of inputs or post-processing of outputs.*

– P6, L22-25: This is an important point, maybe the authors could expand more on this for the non-data assimilation expert?

**We give some more details, especially in terms of linearisation around a given point to approximate the observation operator by its Jacobian.**

*p7 l11: Analytical inversions can also be used on slightly non-linear problems, by linearizing the observation operator around a given reference point using the tangent linear of the observation operator. It formulates as follows:*

$$\mathcal{H}(\mathbf{x}^b + \delta\mathbf{x}) \approx \mathcal{H}(\mathbf{x}^b) + d\mathcal{H}_{\mathbf{x}^b}(\delta\mathbf{x}) = \mathcal{H}(\mathbf{x}^b) + \mathbf{H}_{\mathbf{x}^b}\delta\mathbf{x} \tag{1}$$

*with $\delta\mathbf{x}$ a small deviation from $\mathbf{x}^b$ within a domain where the linear assumption is valid, $d\mathcal{H}_{\mathbf{x}^b}$ the tangent-linear of $\mathcal{H}$ at $\mathbf{x}^b$ and $\mathbf{H}_{\mathbf{x}^b}$ the Jacobian matrix of $\mathcal{H}$ at $\mathbf{x}^b$.*

*Then Eq. (3) can be easily adapted by replacing $(\mathbf{y}^o - \mathbf{H}\mathbf{x}^b)$ by $(\mathbf{y}^o - \mathcal{H}(\mathbf{x}^b))$ and $\mathbf{H}$ by $\mathbf{H}_{\mathbf{x}^b}$.*

*The computation of an analytical inversion faces two main computational limitations. First, the matrix $\mathbf{H}$ representing the observation operator $\mathcal{H}$ must be built explicitly. This can be done either column by column, in the so-called response function method, or row by row, in the so-called footprint method. The two approaches require dim($\mathcal{X}$), the dimension of the target space and dim($\mathcal{Y}$), the dimension of the observation space, independent simulations respectively. In the response function method, each column is built by computing $\{d\mathcal{H}_{\mathbf{x}^b}(\delta\mathbf{x}_i) \setminus \forall \delta\mathbf{x}_i \in \mathcal{B}_\mathcal{X}\}$ with $\mathcal{B}_\mathcal{X}$ the canonical basis of the target space. For a given increment $\delta\mathbf{x}_i$, the corresponding column gives the sensitivity of observations to changes in the corresponding component of the target space. In the footprint method, each row is built by computing $\{\mathcal{H}^*_{\mathbf{x}^b}(\delta\mathbf{y}_i) \setminus \forall \delta\mathbf{y}_i \in \mathcal{B}_\mathcal{Y}\}$ with $\mathcal{B}_\mathcal{Y}$ the canonical basis of the observation space. For a given perturbation of $\delta\mathbf{y}_i$ of a component of the observation vector, the corresponding row of $\mathbf{H}$ gives the sensitivity of the inputs to that perturbation.*

– P7, L8: Are control space and target space the same thing or different? Please be consistent (I would recommend control space as it is more generally used in atmospheric DA) or explain the differences.

**We think that the use of "control vector" can be misleading in some context and prefer "target vector" which is also widely used in the community. We have checked for the consistency of the use of target v control vector over all the manuscript.**

– P7, L12: This part of the sentence is not very clear. Replace "... problems such as the variational and the ensemble Kalman filter methods which are described below."

**We have replaced by:**

*For this reason, smart adaptations of the inversion framework (including approximations and numerical solvers) are often necessary to tackle problems even when they are linear; in the following, we choose to elaborate on some of the most frequent approaches used in the atmospheric inversion community: the variational approach and one ensemble method, the Ensemble Square Root Filter.*

– P7, L18: "limited non-linearity": This is not necessarily true, or it is misleading. EnKFs are frequently applied successfully on chemical transport models which can be significantly non-linear systems in the polluted boundary layer for example.

**Indeed, the ensemble methods can in principle handle non-linear systems, although we found in a CO2-application of the EnSRF that non-linearity in the observation operator quickly degrades the analysis (Tolk et al., 2011). We have revised the text**.

*In the current version, only the EnSRF approach is implemented in the CIF. One should note that the EnSRF, as a direct approximation of the analytical solution, can be very sensitive to non-linearity in the observation operator (e.g., Tolk et al., 2011). It can generally cope only with slight non-linearity over the assimilation window, thus, the assimilation window length has to be chosen appropriately, contrary to other ensemble methods which are usually not sensitive to non-linearity.*

– P7, L25: "running computation window": I believe the authors refer to what is commonly called assimilation window? (Note that inversion is part of data assimilation techniques, so it is fine to call it data assimilation window in the inversion framework). Please be consistent in the terminology in this paragraph. "Smaller", smaller than what? I understand that for GHG inversions especially CO2 you need long window given the observation network and the CO2 atmospheric lifetime, but this is not general to all atmospheric composition inversion.

**We have clarified the terminology. For a given period of interest, EnSRF computes smaller running assimilation windows. The length of the assimilation windows can vary depending on the scale, on the species of interest, the process of interest, the observations used, etc. All the ensemble methods section has been rewritten.**

- P7, L29-31: "for very dense...may not be sufficient": This is not necessarily true, or the statement is misleading. This also needs to be proven. If the reference exists please add it. One can foresee that with a better coverage and higher satellite sensitivity towards the surface the window length could be reduced hence the number of observations to go through sequentially is reduced. Please correct or remove this statement or justify this more clearly.

**We agree with the reviewer that this statement was too vague. The assimilation window must be shortened as much as possible to limit the number of observations, but too short windows generate their own problems of independence of windows and propagation of information between windows. We have clarified to:**

*for very dense observations, such as datasets from new-generation high-resolution satellites, the sequential assimilation of observations may not be sufficient, or at least methods may be needed to make the observation errors between sequential assimilation windows independent, for example by applying a whitening transformation to the observations to form a new set with uncorrelated errors as suggested by Tippet et al., 2003. The challenge is exacerbated for long-lived species such as CO2, for which assimilation windows must be long enough to maintain the propagation of information on the fluxes on long distances through transport. Propagating an covariance matrix from assimilation windows to assimilation windows as accurate as possible could in principle limit the later issue, as suggested in Kang et al., 2011 and 2012, but could still prove hard to apply in very high resolution problems.*

- P8, L11: "under-estimating": This is not necessarily true. Small ensemble size can over and underestimate the uncertainty. More importantly they introduce spuriousness. Please clarified the statement.

**We have clarified the statement. Small ensembles can cause the posterior ensemble to collapse, i.e., the posterior distribution is dominated by one or a very small number of members, which do not allow for a reliable assessment of the posterior uncertainties (Morzfeld et al., 2017); moreover, small ensembles introduce spuriousness in the posterior uncertainties, with irrealistic correlations being artificially generated.**

*p10: By using random sampling, ensemble methods are able to approximate large dimensional matrices at a reduced cost without using the adjoint of the observation operator (see variational inversion below) that can be challenging to implement. Small ensembles generally cause the posterior ensemble to collapse, i.e., the posterior distribution is dominated by one or a very small number of members, which does not allow for a reliable assessment of the posterior uncertainties (Morzfeld et al., 2017); moreover, small ensembles introduce spuriousness in the posterior uncertainties, with irrealistic correlations being artificially generated.. In the EnSRF, small ensembles rather cause a misrepresentation of uncertainty structures, which limits the accuracy of the computed solution, and may require fixes as described in, e.g., Bocquet (2011). In any case, the level of approximation necessary for this approach to work is strongly different for each problem, which requires preliminary studies before consistent application. In particular, the so-called localization of the ensemble can be used to improve the consistency of the inversion outputs (e.g., Zupanski et al., 2007; Babenhauserheide et al., 2015).*

*In the current version, only the EnSRF approach is implemented in the CIF. One should note that the EnSRF, as a direct approximation of the analytical solution, can be very sensitive to non-linearity in the observation operator (e.g., Tolk*

*et al., 2011). It can generally cope only with slight non-linearity over the assimilation window, thus, the assimilation window length has to be chosen appropriately, contrary to other ensemble methods which are usually not sensitive to non-linearity.*

– P8, L13-14: Could you possibly provide few examples of different problems having different approximations?

**We give two examples of past studies who discussed the use of the localization of the Monte Carlo ensemble to help the filter keeping consistent results.**

– P8 L15-28: I recommend revising this introduction of the variational method. Add a sentence to present the cost function. Put the equation right after. Explain in few sentences that the aim is to minimize this cost function and that to achieve this the gradient is calculated. Then put the gradient formula.

**We have re-arranged the variational section to clarify it. See track-change for detailed modifications. In particular, we give details on how the posterior uncertainties are computed in variational inversions in the new version of the manuscript.**

– P8, L25: I am not sure this is true. If the state vector is large, i.e. increasing resolution or augmenting the control variables, or increasing the number of observations, the minimization is expected to be slower.

**The limiting factor in the analytical inversion approach is the size of H which cannot be broken down (unlike B). The variational approach still has limitations but memory is unlikely to be a problem since B can be broken down, only the computational time could be a limiting factor.**

**Technically, crazily huge target vector will impede the capability of variational inversions to be computed, if only due to memory limitations, and convergence time. However, these limits come for much bigger problems than for analytical and ensemble methods. Also, the minimization algorithms such as CONGRAD and M1QN3 are extremely efficient and manage a relatively good level of convergence with only a few tens of iterations even in very large dimensional problems (when correctly set up). For very large inversion problems (especially in the case of non-linearity), variational inversions still need thorough set up to guarantee the convergence.**

– P12, Equation 12: Please, be explicit about what the index represents. "... of elementary interchangeable N transformations ..."

**We now give more details about this equation. This generalization of the decomposition of the observation operators allow us to simplify the configuration and development of inversions. In particular, the development of new functions is well identified and makes the development of the adjoint simpler.**

*p15 l15: In such a formalism, all intermediate transformations have the same conceptual level in the code. They are functions ranging from spatial reprojection, to temporal interpolations, to more complex operations such as the reconstruction of satellite total columns from concentrations simulated at individual levels in the transport model. In the CIF, all these transformations have the same input and output structure and, thus, their order can be changed seamlessly*

*to execute a given configuration. Please note that the commutative property of elementary transformations as pieces of code does not guarantee the commutative property of the corresponding physical operators.*

– P13, L10: It is a bit of pity that air quality application is only mentioned here. I think a system such as CIF could greatly benefit the air quality community as much as the GHG community, which are getting more connected lately.

**We agree. As mentioned above, the CIF is being developed by members of the GHG community primarily. However, air quality applications are also included in the purposes of the CIF, as e.g., the chemistry transport CHIMERE is implemented in the CIF with complex chemical schemes. AQ applications are now mentioned from the introduction.**

– P13, L13: I am not sure what the authors meant by spatial gradient as observations? Please clarified or remove.

**Some inversion studies assimilate differences between concentrations at different locations in the observation vector (Breon et al. 2015), instead of individual observations. This approach is meant to limit the influence of background concentrations in the inversion of local fluxes.**

*p15 l7: authors implemented differences between observation sites and time in the observation vector instead of observations from individual sites in order to focus on spatial/temporal gradients, thus allowing to limit the influence of background concentrations in the computation of local*

– P13, L14-15: Maybe the authors could mention here that the main idea behind "super-obing" is to lower the discrepancies of the representativeness between model and observations.

**We thank the reviewer for helping to clarify our text. We have added statements about the representativeness of the observations**

*p16 l4: In the case of super-observations (satellites retrievals, images, spatial gradients, etc.) in the observation vector, it is often necessary to combine multiple simulated point observations in given grid cells and time stamps into a single super-observation, to limit redundant observations, hence the size of the observation vector.*

– P13, L15-16: "This is the case for...time and locations," This sentence is very unclear. Please rephrase or explain better.

**We agree with the reviewer. We have split and clarified our statement.**

*p16 l8: Super-observations are commonly used in the case for satellite observations being compared to all the model levels above a given location; concentration gradients comparing observations at different time and locations (see e.g., Bréon et al., 2015; Staufer et al., 2016) are another example of observation aggregation to reduce representativeness errors. isotopic ratios are also super-observations as10they require to simulate separate isotopologues and recombine them after the simulation (as done in e.g.,van der Velde et al., 2018; Peters et al., 2018)*

– P15, L21-31: What about satellite observations? I think the authors should add statement about this.

**At the moment (in a version of the CIF more advanced than the one described in the present paper), satellite observations are included in the observation operator, but the user must format original files manually to a standard observation format readable by the CIF, with column, averaging kernels, etc. in a specific form. In the future, there will be a need to allow raw satellite formats automatically in the CIF to be ingested without manual formatting.**

*p19 l1: satellite products, in particular the formatting of averaging kernels and other metadata, should also be included in the CIF in the near future as they play a growing role in the community*

– P17, L11: Those studies use Gaussian plume technique but do not provide a continuous inversion constrain globally. Please correct the statement as it makes the reader believes that the Gaussian plume technique would allow a global inversion similar to a CTM based inversion.

**We fully agree with the reviewer and have clarified the sentence.**

*p20 l22: to the larger scales with satellite measurements to optimize individual clusters of industrial or urban emissions*

– P19, L28: 10000m is 4 times the size of the domain here. No data assimilation system would use such long length scale. Is this on purpose? If yes, please justify.

**The purpose of this set-up is to demonstrate that very large correlation lengths are equivalent to having only one uniform scaling factor. In the example set-up we had, the high number of observation sites counter-balanced the large correlation lengths. We changed our set-up, with a smaller domain and smaller number of observation sites to really demonstrate the equivalence of large correlation lengths and one single scaling factor for the whole domain. The corresponding figures are shown in Supplement now.**

– P20, L16, Fig 5 and similar other figures. With all the fluxes and fluxes differences plots it would be better to use python colour maps that are diverging and centred around 0. This will improve the readability of the figures.

**We have used centered color scales for the uncertainty matrix as plotted now. But as fluxes are not centered, we find that our color scales are the best compromise for readability.**

– P20, L20: I am unsure if I understand correctly but I believe the authors are speaking about iterations in the variational sense and members in the EnKF sense. If yes, please have few sentences to explain what the word 'simulation' means. Alternatively, I would replace the word 'simulations' by'iterations or members.'

**The reviewer understood correctly. We clarified. We use this measure in the x-axis to be comparable between variational and ensemble methods, as one iteration of a variational algorithm requires two simulations (one forward and one adjoint). Details are given about the number of simulations in Sect. 4.2.2**

*p24 l30: More precisely, users and developers of adjoint transport models choose the number of forward re-computations to be carried out, based on a space-speed trade off: by saving all forward intermediate states, the adjoint is as costly as the forward, but the disk space burden can be extremely challenging to manage, thus increasing the overall computation time in return.*

– P20, L32: 10000m is 4 times the size of the domain here. See the related comment above. Such long length scales would provide very uniform structure on the increments. Structure similar to the 500m increments is seen on the result using the 10000m length scale. I expect there is typo through the text a 0 should be removed to 1000m?

**This was indeed 10000m, but the high number of observation sites counter-balanced the long correlation lengths. We have changed the set-up with less observation sites to have very uniform increments when having very large correlation scales. This is a simple test to demonstrate the consistency of the definition of the correlations in the system. To make this test clearer, we now prescribe a correlation of 200 km, hence 100 times the size of the domain.**

– P20,L33-P21,L2: This look to me more like a bug rather noise as vertical lines or "chessboard" like pattern appear in the increments results. Such pattern also occurs in the analytical solution. This not looking like sampling noise to me. I think this points out possible problems in the implementation of the EnKF and of the systems other than the variational methods. It is not entirely clear if the system uses the ensemble information to derive jacobians for the calculation of H and perform a minimization? In this case this inversion cannot be called EnKF. This needs to be revised or diagnosed or explained.

**Thanks for pointing this out. There was a small difference in the definition of the B matrix between analytical and ensemble methods on one side, and variational inversions in the other site. We have removed it and have correspondingly re-computed the inversions.**

**1.3 Technical comments**

– P2, L27-29: an order in all those references, chronological or alphabetical.

**We have re-ordered those references chronologically**

– P7, L16: replace "... linear, simple cases." by "... linear and simple cases."

**Corrected**

– P7, L19: change "characterized" to "characterize".

**Corrected**

– P9, L13: Chevallier et al., 2005: put this reference to the previous parenthesis for clarity.

**Corrected**

– P10, L5: There is a colon here but nothing after. Should we expect equations?

**An equation was missing in the submitted version of the manuscript. We detail how posterior matrices can be deduced in variational inversions: either with Monte Carlo experiments or with truncated eigen vectors and values of the Hessian matrix**

*p12 l8: The approximation of the posterior uncertainty matrix* $\mathbf{A}$ *in the case of CONGRAD reads as follows:*

$$\mathbf{A} = Hess(J)_{\mathbf{x}^a}^{-1} \approx \mathbf{V}_{\mathbf{x}^a}^{\mathrm{T}} \mathbf{\Lambda}_{\mathbf{x}^a}^{-1} \mathbf{V}_{\mathbf{x}^a} \tag{2}$$

*with* $\mathbf{V}_{\mathbf{x}^a}$ *the dominant eigenvectors of the Hessian matrix at the point* $\mathbf{x}^a$ *and* $\mathbf{\Lambda}_{\mathbf{x}^a}$ *the matrix of the dominant eigenvalues of the Hessian matrix. Please note that the dominant eigenvalues of the Hessian matrix correspond to components with low posterior uncertainties in* $\mathbf{A}$.

*Another approach to quantify the posterior uncertainty matrix* $\mathbf{A}$, *valid for both linear and non-linear cases, is to carry out a Monte Carlo ensemble of independent inversions with sampled prior vectors from the prior distribution* $\mathcal{N}(\mathbf{x}^b, \mathbf{B})$ *(e.g., ?). An ensemble of posterior vectors are inferred and used to compute the posterior matrix as follows:*

$$\mathbf{A} \approx \frac{1}{N-1} (\mathbf{x}_1^{\mathrm{a}} - \mathbf{x}_{ref}^{\mathrm{a}}, \ \mathbf{x}_2^{\mathrm{a}} - \mathbf{x}_{ref}^{\mathrm{a}}, \ \dots \ \mathbf{x}_N^{\mathrm{a}} - \mathbf{x}_{\mathrm{ref}}^{\mathrm{a}}) \cdot (\mathbf{x}_1^{\mathrm{a}} - \mathbf{x}_{ref}^{\mathrm{a}}, \ \mathbf{x}_2^{\mathrm{a}} - \mathbf{x}_{ref}^{\mathrm{a}}, \ \dots \ \mathbf{x}_N^{\mathrm{a}} - \mathbf{x}^T) \tag{3}$$

*with* $N$ *the size of the Monte Carlo ensemble,* $\mathbf{x}_i^a$ *the posterior vector corresponding to the prior* $\mathbf{x}_i^b$ *of the Monte Carlo ensemble and* $\mathbf{x}_{ref}^a$ *the average over sampled posterior vectors.*

– P15, L18: I understand what the authors means but this need to be reformulated. "Meteos" is not really correct English.

**This referred to the keyword used in the CIF, which is abbreviated for the sake of readability; we replaced the word in the body of the manuscript by "meteo-data".**

– P19, L3-10: Maybe add the equations in the text where they are mentioned.

**We re-aranged this section accordingly.**

– P22, L6: This is not a fair statement considering on how the "EnKF" has been implemented here. It doesn't seem to be strictly an EnKF as the sequential assimilation is not performed. Consider changing or removing this in the conclusion. Also, this is not the scope of GMD paper here to compare methods but to describe and showcase model/software developments for geoscience.

**We agree that a GMD "description paper" should not go too far in interpreting demonstration results. We have reformulated and reduced the comparison of methods. See track-change for modifications in the result section.**

– P22, L10-12: The authors could add some sentences about the challenges in implementing actual CTMs in the CIF framework. For example, among many other challenges, I/O is a limiting factor for data assimilation especially with CTMs.

**We agree that I/O can be a limiting factor. With its very general formalism, the CIF may not be able to accommodate very specific tweaks necessary for some models and some very large applications. However, using**

**very efficient numerical libraries in python (which uses C in the background), one should be able to keep up the pace with systems with compiled codes (Fortran mainly in the community). We have added a paragraph in the conclusion to discuss this issue:**

*The next step of the CIF is the implementation of a large variety of CTMs. The implementation of new CTMs already interfaced with other inversion systems should not bring particular conceptual challenges as all interface operations are already written in their original inversion system; in most cases, re-arranging existing routines is sufficient to interface a model to the CIF. One particular challenge concerns I/O optimizations: the generation of inputs and the processing of outputs can be time consuming and in some very heavy applications requires specific numerical and coding optimizations. The very general formalism of the CIF may hamper the ability of applying these particular optimizations for some models. Best efforts will have to be deployed to take full advantage of advanced I/O and data manipulation libraries in python to limit this weakness.*

**2    Referee #2: Peter Rayner**

**2.1    General Comments**

This paper introduces a software system to facilitate ensembles of inverse calculations. The likely focus is atmospheric inversions in which model inputs such as surface fluxes are inferred from combinations of prior information and observations. Such a system is clearly useful since ensembles based on choices of ingredients (such as models or input covariances) provide a better basis for both the means and uncertainties of posterior estimates. the paper is certainly within scope for GMD since it describes a system which might well be widely used in the future. It is also well written though some final proofreading, especially around pluralisation, will be needed.

I note I am not reviewing the software itself. There is only one major critique of the paper. I think the benefits of the system are slightly over-sold (but what software re-lease doesn't do this). I believe, for example. that CIF doesn't provide a framework for comparing estimates where the target spaces differ such as ENKF with short windows vs long-term mean approaches. That's an intellectual/mathematical task which one could instantiate in software. This would probably involve classes representing PDFs which might be internally represented by ensembles or parameters and functions then defining transformations among these PDFs. That quibble aside there is no doubt the system will help such comparisons.

**We agree that the CIF does not provide a plug'n'play solution to the problem of comparing totally different inversion set-ups in a statistically consistent way. Doing so would require further mathematical considerations that may prove necessary in the future to compare methods/models that cannot be computed on, e.g., exactly the same inversion window (i.e., when comparing inversion results for a given region, one computed with a regional model and the other with a global one). However, having a consistent framework with comparable configuration parameters and identical output format will definitely help to develop such a comparison.**

On the positive side the ability to configure the computational pipeline is a marvellous innovation. Building our own, less ambitious system, we have struggled to make it agnostic about computation.

Perhaps more important than supporting intercomparisons, the tools provided as part of the system will lower the entry barrier for new groups entering the field. This was a surprising spin-off from the TRANSCOM effort 20 years ago. Most systems since then have been too closely tied to a given model for universal use so CIF can provide a major boost to the field. The fact that it is well documented will only help this.

There is one aspect, though, that seems seriously deficient. If I read correctly, methods for calculating uncertainty in the Gaussian posterior are not provided. One important role of a new framework like CIF is to educate, to encourage good practice. That includes calculating and engaging with the uncertainty in the results. Certainly any of the analytic methods under-estimate this term but it is still true that under-sampling of concentration is a major source of uncertainty for inversions at all scales. I request that the authors at least detail a road map for including posterior uncertainty.

**We agree: this was missing. Of course the posterior uncertainty calculation depends on the approach used (analytical, CONGRAD, M1QN3, or EnKF), as well as how the error matrices are computed in the system. We have implemented the computation of the posterior error matrices for all methods and have re-arranged the manuscript to demonstrate the capability to provide uncertainty reduction and posterior error matrices. In particular, posterior matrices are shown in the Result section.**

**2.2 Specific Comments**

- P7 bottom this correlation of errors is a common critique of ensemble methods but I've rarely seen it implemented in a non-ensemble method either and it's perfectly possible to do in ensemble methods, one just adds some extra state variables carrying observational corrections and build the correct autocorrelations into them instead

  **We agree most variational inversions assume diagonal observation error matrices, or block diagonals, which is somehow equivalent to the running assimilation window of the Kalman Filter.**

- P8 L22saying "the maximum probability" and "the mode" is tautological I think

  **We have re-arranged the variational section following comments from reviewer #1 and have removed this tautology.**

- P9 congratulations for explicitly writing out the regularisation/reduction step which is so often assumed. It is called pre-conditioning so often that you should add the term for clarity

  **We now mention the term pre-conditioning for consistency with the terminology used in the community**

- P9 it's worth commenting that leading eigen-vectors of the Hessian correspond to the low uncertainty parts of the solution which might or might not be what you want

  **Thank you for pointing out this aspect. We now give details on the formula used to deduce the posterior matrix from the eigen vector of the Hessian and mention this.**

*p12 l8: The approximation of the posterior uncertainty matrix $\mathbf{A}$ in the case of CONGRAD reads as follows:*

$$\mathbf{A} = Hess(J)_{\mathbf{x}^a}^{-1} \approx \mathbf{V}_{\mathbf{x}^a}^{T} \mathbf{\Lambda}_{\mathbf{x}^a}^{-1} \mathbf{V}_{\mathbf{x}^a} \tag{4}$$

*with $\mathbf{V}_{\mathbf{x}^a}$ the dominant eigenvectors of the Hessian matrix at the point $\mathbf{x}^a$ and $\mathbf{\Lambda}_{\mathbf{x}^a}$ the matrix of the dominant eigenvalues of the Hessian matrix. Please note that the dominant eigenvalues of the Hessian matrix correspond to components with low posterior uncertainties in $\mathbf{A}$.*

*Another approach to quantify the posterior uncertainty matrix $\mathbf{A}$, valid for both linear and non-linear cases, is to carry out a Monte Carlo ensemble of independent inversions with sampled prior vectors from the prior distribution $\mathcal{N}(\mathbf{x}^b, \mathbf{B})$ (e.g., ?). An ensemble of posterior vectors are inferred and used to compute the posterior matrix as follows:*

$$\mathbf{A} \approx \frac{1}{N-1} (\mathbf{x}_1^a - \mathbf{x}_{ref}^a, \; \mathbf{x}_2^a - \mathbf{x}_{ref}^a, \; \dots \mathbf{x}_N^a - \mathbf{x}_{\text{ref}}^a) \cdot (\mathbf{x}_1^a - \mathbf{x}_{ref}^a, \; \mathbf{x}_2^a - \mathbf{x}_{ref}^a, \; \dots \mathbf{x}_N^a - \mathbf{x}^T) \tag{5}$$

*with $N$ the size of the Monte Carlo ensemble, $\mathbf{x}_i^a$ the posterior vector corresponding to the prior $\mathbf{x}_i^b$ of the Monte Carlo ensemble and $\mathbf{x}_{ref}^a$ the average over sampled posterior vectors.*

– P11 citing the Bousserez and Henze paper is a good advertisement for your approach, you might want to say in the abstract or intro that many new methods are hybrid and this combined framework simplifies implementing such methods

**We insist on the advantage for modern hybrid methods of such a flexible structure. The decomposition of elementary operations makes it easier to recompose them in various ways. Besides, the CIF allows new methods to be tested with multiple models, hence inquiring their strengths and weaknesses in various contexts.**

*p25 l14: With modern inversion methods moving towards an hybrid paradigm of variational and ensemble methods, the flexibility of the CIF will be a valuable asset as abstract methods can be easily mixed with each other.*

– P15 a technical question, how do you handle target variables which aren't needed at various stages, e.g. bias corrections in observations which aren't inputs to the CTM? the pipe lining structure doesn't lend itself well to this problem.

**Fig 2 and Eq. 11-12-13 may give the impression that all elementary transformations are computed in a serialized way. One should keep in mind that $\mathbf{x}$ in Figure 2 includes all target variables, not only those regarding fluxes (as assumed in the figure). Therefore, if some variables are related to observation biases, they are kept identical along all transformations before and after the models, and they are only used at the last step when reconstructing the simulated observation vector equivalent. We have modified the figure to highlight this aspect. The text describing the figure has been extended:**

*p16 l13: The case of Fig. 2 also include background concentrations in the target vector: a background is often used to fix some biases in initial and lateral concentrations in limited-area models, and in observations (mostly satellites); the background variables are processed at the very end of the pipe when re-constructing the observations vector.*

*The mathematical formalism of Eq. (15) and (16) suggests that transformations are necessarily computed in a serialized way, thus limiting applications to simple target variables upstream the transport model. However, each elementary transformation handles components of the inputs it is concerned with, leaving the rest identical and forwarding it to later transformations. Typically, it does not actually limit applications to simple target variables upstream the CTM. For instance, in the case of target variables optimizing biases in the observations, the corresponding components of the target vector $\mathbf{x}$ are forwarded unchanged by all transformations in Fig. 2 until the very last operation, where they are used for the final comparison to the observation vector.*

– P18 I think Eq. 17, do you consider sources as points or area integrals, if so do you need to integrate over the area of the source, especially for nearby observations?

**The formulas used in Eq. 16 and 17 are based on point sources. The word "gridded" in the text is therefore misleading. We have clarified this statement by:**

*p21 l18: we use a grid of point surface fluxes to simulate concentrations using the Gaussian plume equation.*

– P20 it's worth adding that the amount of intermediate calculation needed is at user discretion; a space-speed trade-off. CMAQ, for example needs almost none, having check pointed what it needs on the forward run it had to do anyway at the start of the iteration.

**We thank the reviewer for highlighting this aspect. We have extended the paragraph as follows to mention this aspect:**

[revised manuscript text omitted]
}^b)^T \mathbf{B}^{-1}(\mathbf{x} - \mathbf{x}^b) + \frac{1}{2}(\mathbf{y}^o - \mathcal{H}(\mathbf{x}))^T \mathbf{R}^{-1}(\mathbf{y}^o - \mathcal{H}(\mathbf{x})) \tag{6}$$

5     ~~The variational formulation does not require calculation of complex matrix products and inversions, contrary to the analytical inversion, and is thus not limited by vector dimensions. Still, the inverses of the uncertainty matrices $\mathbf{B}$ and $\mathbf{R}$ need to be computed, potentially prohibiting the use of very large and/or complex general matrices; this challenge is often overcome by reducing $\mathbf{B}$ and $\mathbf{R}$ to manageable combinations of simple matrices (e.g., Kronecker products of simple shape covariance matrices; see Sect. 2.3.1).~~

10     In variational inversions, the minimum of the cost function in Eq. (6) is numerically estimated iteratively using quasi-Newtonian algorithms based on the gradient of the cost function:

$$\nabla J_{\mathbf{x}} = \mathbf{B}^{-1} \cdot (\mathbf{x} - \mathbf{x}^b) + \mathcal{H}^* \left( \mathbf{R}^{-1} \cdot (\mathbf{y}^o - \mathcal{H}(\mathbf{x})) \right) \tag{7}$$

    Quasi-Newtonian methods are a group of algorithms designed to compute the minimum of a function iteratively. It should be noted that in high-dimension problems, it can take a very large number of iterations

15     to reach the minimum of the cost function $J$, forcing the user to stop the algorithm before convergence, thus reaching only an approximation of $\mathbf{x}^a$; in addition, iterative algorithms can reach local minima without ever reaching the global minimum, making it essential to thoroughly verify variational inversion results; this can happen in  non-linear cases, but also, due to numerical artefacts, in linear cases (some points in the cost function have gradients so close to zero that the algorithm sees them as convergence points, whereas

20     the  unique global minimum is somewhere else). In the community, examples of quasi-Newtonian algorithms commonly used are  the Broyden–Fletcher–Goldfarb–Shanno (BFGS) algorithm (Zheng et al., 2018; Bousserez et al., 2015), M1QN3 (Gilbert and Lemaréchal, 1989), and the CONGRAD algorithm (applicable only to linear or linearized problems; Fisher, 1998; Chevallier et al.,

25     based on the Lanczos method, which iterates to find the eigenvalues and eigenvectors of the Hessian matrix, which is then used (in a single step) to calculate the analysis vector, $\mathbf{x}^a$. In general, quasi-Newtonian methods require an initial regularization, or "pre-conditioning" of $\mathbf{x}$, the vector to be optimized, for better efficiency. In atmospheric inversions, such a regularization is generally made by optimizing $\chi = \mathbf{B}^{-1/2} \cdot (\mathbf{x} - \mathbf{x}^b)$ instead of $\mathbf{x}$; we note $\mathfrak{A}$ the regularization space: $\chi \in \mathfrak{A}$. This transformation translates in Eq. (6) and (7) as follows:

$$\nabla J_\chi = \chi + \mathbf{B}^{1/2}.\mathcal{H}^*\mathbf{R}^{-1}.(\mathbf{y}^o - \mathcal{H}(\mathbf{x}))\begin{cases} J_\chi &= \frac{1}{2}\chi^{\mathrm{T}}\chi + \frac{1}{2}(\mathbf{y}^o - \mathcal{H}(\mathbf{B}^{1/2}.\chi + \mathbf{x}^b))^{\mathrm{T}}\mathbf{R}^{-1}(\mathbf{y}^o - \mathcal{H}(\mathbf{B}^{1/2}.\chi + \mathbf{x}^b)) \\ \nabla J_\chi &= \chi + \mathbf{B}^{1/2}.\mathcal{H}^*\left(\mathbf{R}^{-1}.(\mathbf{y}^o - \mathcal{H}(\mathbf{B}^{1/2}.\chi + \mathbf{x}^b))\right) \end{cases}$$
$$(8)$$

Solving Eq.  (6) and (7) in the target vector space or Eq.  (8) in the regularization space is mathe-
5  matically fully equivalent, but the solution in the regularization space is often reached in fewer iterations.
Moreover, in the regularization space, one can force the algorithm to focus on the main modes of the target
vector space by filtering the smallest eigenvalues of the matrix $\mathbf{B}$. This reduces the dimension of $\chi$ and
accelerates further the rate of convergence, although the solution of the reduced problem is only an approxi-
mation of the solution of the full problem. In the following we thus prefer calling the "regularization space"
10  the "reduction space". The link between the two can be written as follows:

$$\begin{aligned} \chi_{\text{full}} &= \mathbf{Q}\boldsymbol{\Lambda}^{-1/2} & (\mathbf{x} - \mathbf{x}^b) \\ \chi_{\text{reduced}} &= \mathbf{Q}'\boldsymbol{\Lambda}'^{-1/2} & (\mathbf{x} - \mathbf{x}^b) \end{aligned}$$
$$(9)$$

with  $\mathbf{B}^{1/2} = \mathbf{Q}\boldsymbol{\Lambda}^{1/2}\mathbf{Q}^{\mathrm{T}}$, $\mathbf{Q}$ and $\boldsymbol{\Lambda}$ being the matrices of the eigenvector and the ma-
trix of the corresponding eigenvalues of the matrix $\mathbf{B}$. $\mathbf{Q}'$ and $\boldsymbol{\Lambda}'$ are the reduced matrices of eigenvalues
and eigenvectors with a given number of dominant eigenvalues.

15  Overall, variational inversions are a numerical approximation to the solution of the inversion problem:
they involve the gradient of the cost function in Eq. (7) and require to run forward and adjoint simulations
iteratively (e.g., Meirink et al., 2008; Bergamaschi et al., 2010; Houweling et al., 2016, 2014; Fortems-Cheiney et al., 2021; Chevallier et al., 201

The variational formulation does not require calculation of complex matrix products and inversions,
20  contrary to the analytical inversion, and is thus not limited by vector dimensions. Still, the inverses of the
uncertainty matrices $\mathbf{B}$ and $\mathbf{R}$ need to be computed, potentially prohibiting the use of very large and/or
complex general matrices; this challenge is often overcome by reducing $\mathbf{B}$ and $\mathbf{R}$ to manageable combinations
of simple matrices (e.g., Kronecker products of simple shape covariance matrices; see Sect. 2.3.1).
When the observation operator is linear, the posterior uncertainty matrix $\mathbf{A}$ is equal to the inverse of
25  the Hessian matrix at the minimum of the cost function. In most cases the Hessian cannot be computed
explicitly, because of memory limitations, which is a major drawback of variational inversions. But some
variational algorithms such as CONGRAD provide a coarse approximation of the Hessian: in the case of
CONGRAD based on the Lanczos method, leading eigenvectors of the Hessian can be computed, together

with their eigenvalues (Fisher, 1998). The approximation of the posterior uncertainty matrix $\mathbf{A}$ in the case of CONGRAD reads as follows:

$$\mathbf{A} = Hess(J)^{-1}_{\mathbf{x}^a} \approx \mathbf{V}^{T}_{\mathbf{x}^a} \mathbf{\Lambda}^{-1}_{\mathbf{x}^a} \mathbf{V}_{\mathbf{x}^a} \tag{10}$$

with $\mathbf{V}_{\mathbf{x}^a}$ the dominant eigenvectors of the Hessian matrix at the point $\mathbf{x}^a$ and $\mathbf{\Lambda}_{\mathbf{x}^a}$ the matrix of the dominant eigenvalues of the Hessian matrix. Please note that the dominant eigenvalues of the Hessian matrix correspond to components with low posterior uncertainties in $\mathbf{A}$.

Another approach to quantify the posterior uncertainty matrix $\mathbf{A}$, valid for both linear and non-linear cases, is to carry out a Monte Carlo ensemble of independent inversions with sampled prior vectors from the prior distribution $\mathcal{N}(\mathbf{x}^b, \mathbf{B})$ (e.g., Liu et al., 2017). An ensemble of posterior vectors are inferred and used to compute the posterior matrix as follows:

$$\mathbf{A} \approx \frac{1}{N-1} \left( \mathbf{x}^a_1 - \mathbf{x}^a_{ref}, \, \mathbf{x}^a_2 - \mathbf{x}^a_{ref}, \, \dots \, \mathbf{x}^a_N - \mathbf{x}^a_{ref} \right) \cdot \left( \mathbf{x}^a_1 - \mathbf{x}^a_{ref}, \, \mathbf{x}^a_2 - \mathbf{x}^a_{ref}, \, \dots \, \mathbf{x}^a_N - \mathbf{x}^T \right) \
[revised manuscript text omitted]